# Age-related immune response heterogeneity to SARS-CoV-2 vaccine BNT162b2

Dami A. Collier[1,2,3,29], Isabella A. T. M. Ferreira[1,2,29], Prasanti Kotagiri[1,2,29], Rawlings P. Datir[1,2,3,29], Eleanor Y. Lim[2,29], Emma Touizer[3], Bo Meng[1,2], Adam Abdullahi[1], The CITIID-NIHR BioResource COVID-19 Collaboration*, Anne Elmer[4,5], Nathalie Kingston[4,5], Barbara Graves[4], Emma Le Gresley[4,5], Daniela Caputo[4,5], Laura Bergamaschi[1], Kenneth G. C. Smith[1,2], John R. Bradley[2,4], Lourdes Ceron-Gutierrez[6], Paulina Cortes-Acevedo[7], Gabriela Barcenas-Morales[7], Michelle A. Linterman[8], Laura E. McCoy[3], Chris Davis[9], Emma Thomson[9], Paul A. Lyons[1,2], Eoin McKinney[1,2✉], Rainer Doffinger[5✉], Mark Wills[1,2✉] & Ravindra K. Gupta[1,2✉]

Although two-dose mRNA vaccination provides excellent protection against SARS-CoV-2, there is little information about vaccine efficacy against variants of concern (VOC) in individuals above eighty years of age[1]. Here we analysed immune responses following vaccination with the BNT162b2 mRNA vaccine[2] in elderly participants and younger healthcare workers. Serum neutralization and levels of binding IgG or IgA after the first vaccine dose were lower in older individuals, with a marked drop in participants over eighty years old. Sera from participants above eighty showed lower neutralization potency against the B.1.1.7 (Alpha), B.1.351 (Beta) and P.1. (Gamma) VOC than against the wild-type virus and were more likely to lack any neutralization against VOC following the first dose. However, following the second dose, neutralization against VOC was detectable regardless of age. The frequency of SARS-CoV-2 spike-specific memory B cells was higher in elderly responders (whose serum showed neutralization activity) than in non-responders after the first dose. Elderly participants showed a clear reduction in somatic hypermutation of class-switched cells. The production of interferon-γ and interleukin-2 by SARS-CoV-2 spike-specific T cells was lower in older participants, and both cytokines were secreted primarily by CD4 T cells. We conclude that the elderly are a high-risk population and that specific measures to boost vaccine responses in this population are warranted, particularly where variants of concern are circulating.

Vaccines designed to elicit protective immune responses remain the key hope for containing the COVID-19 pandemic caused by SARS-CoV-2. In particular, mRNA vaccines have shown excellent efficacy when administered as two doses separated by a three- or four-week gap[2,3]. There is increasing evidence that neutralizing responses are a correlate of protection[4–6]. Few trial data on neutralizing responses or vaccine efficacy in individuals above the age of 80 are available[1]. This is even more pertinent for settings in which a dosing interval of 12–16 weeks or more has been implemented to maximize the administration of first doses[7]. In addition, the emergence of new variants with increased transmissibility[8] and reduced sensitivity to vaccine-elicited antibodies[9], and for which vaccines are less able to prevent infection[10], has raised fears for vulnerable groups in whom the magnitude and quality of immune responses may be suboptimal.

## Neutralization following immunization

We studied 140 participants who had received at least one vaccination (median age 72 years (interquartile range (IQR) 44–83), 51% female; Extended Data Fig. 1). We first validated the use of a pseudotyped virus (PV) system to investigate neutralization, by comparing geometric mean titres (GMTs) between PVs expressing the Wuhan-1 D614G spike (referred to here as wild-type) and a B.1 lineage live virus isolate, using sera isolated from thirteen individuals after two vaccine doses (Extended Data Fig. 2). We observed a high correlation between the two approaches, consistent with other findings[11], and proceeded with the PV system.

We explored the association between age and ability to neutralize virus by plotting the proportion of individuals whose sera produced detectable virus neutralization after the first dose at a given age. This analysis showed a nonlinear relationship with a marked drop around

[1]Cambridge Institute of Therapeutic Immunology and Infectious Disease (CITIID), Cambridge, UK. [2]Department of Medicine, University of Cambridge, Cambridge, UK. [3]Division of Infection and Immunity, University College London, London, UK. [4]NIHR Bioresource, Cambridge, UK. [5]Department of Public Health and Primary Care, School of Clinical Medicine, University of Cambridge, Cambridge Biomedical Campus, Cambridge, UK. [6]Department of Clinical Biochemistry and Immunology, Addenbrookes Hospital, Cambridge, UK. [7]Laboratorio de Inmunología, FES-Cuautitlán, UNAM, Mexico City, Mexico. [8]Immunology Programme, Babraham Institute, Cambridge, UK. [9]MRC Centre for Virus Research, University of Glasgow, Glasgow, UK. [29]These authors contributed equally: Dami A. Collier, Isabella A. T. M. Ferreira, Prasanti Kotagiri, Rawlings Datir, Eleanor Lim. *A list of authors and affiliations appears at the end of the paper. ✉e-mail: efm30@medschl.cam.ac.uk; rd270@medschl.cam.ac.uk; mrw1004@cam.ac.uk; rkg20@cam.ac.uk

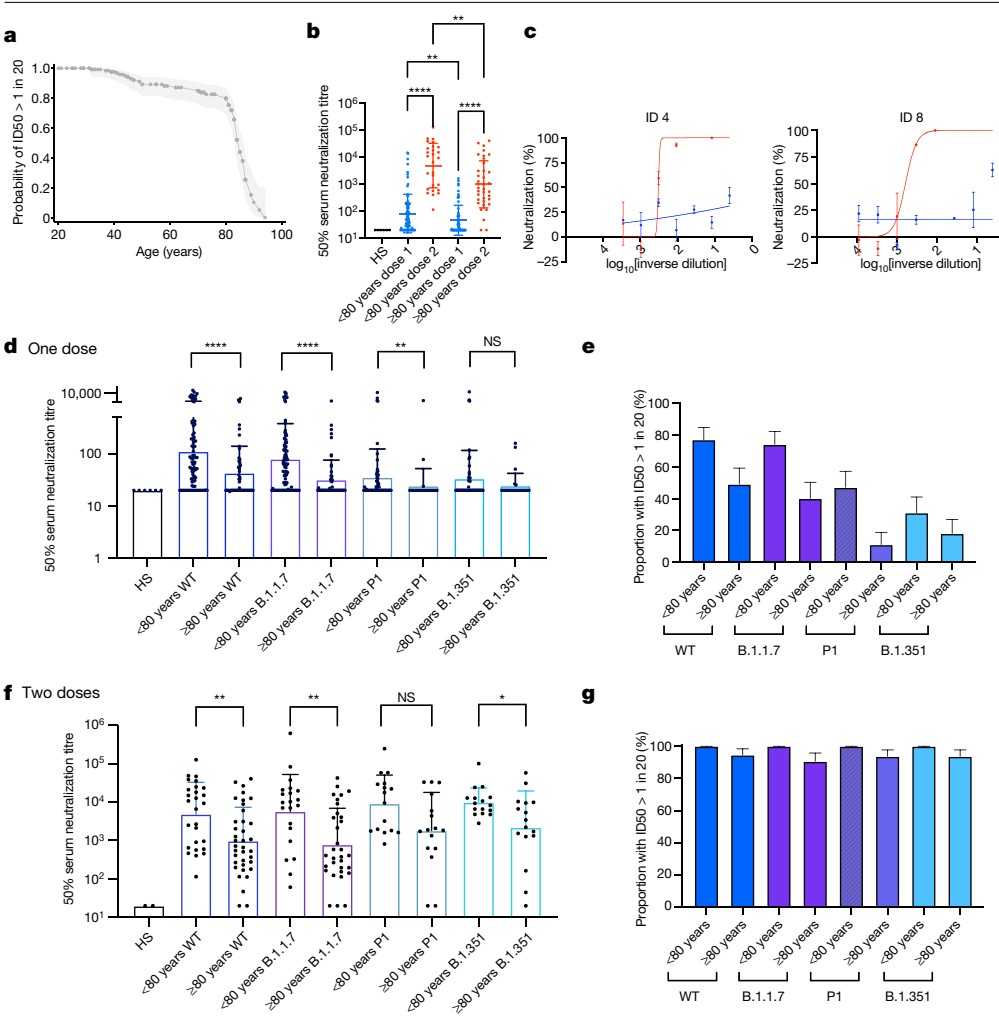

**Fig. 1 | SARS-CoV-2 neutralization by sera from BNT162b2 vaccinated individuals. a**, Proportion of individuals with detectable serum neutralization of PV after the first dose of Pfizer BNT162b2 vaccine by age. Cut-off for serum neutralization is an inhibitory dilution at which 50% inhibition of infection is achieved (ID50) of 20. Shading, 95% CI. **b**, Serum neutralization of PV after dose 1 (blue) and dose 2 (red) by age group (<80 years ($n = 79$), ≥80 years ($n = 59$)). **c**, Neutralization curves for serum from two individuals (ID 4 and ID 8) with lower responses after the first dose (blue) and increased neutralization activity after the second dose (red) of BNT162b2 against pseudovirus expressing wild-type spike protein (D614G). Data shown as mean ± s.e.m. of technical replicates. **d**, **f**, Neutralization of SARS-CoV-2 VOCs by sera after dose 1 (**d**) and dose 2 (**f**) of BNT162b2. **d**, WT, $n = 138$; B.1.1.7, $n = 135$; B.1.351, $n = 82$; P.1, $n = 82$. **f**, WT, $n = 64$; B.1.1.7, $n = 53$; B.1.351, $n = 32$; P.1, $n = 32$. Data shown as GMT ± s.d. **e**, **g**, The proportion of participant vaccine sera with neutralization activity against wild-type and mutant spike proteins after dose 1 (**e**) and dose 2 (**g**) (ID50 > 1 in 20 dilution of sera). GMT ± s.d. are representative of two independent experiments each with two technical repeats. Mann–Whitney test was used for unpaired comparisons and Wilcoxon matched-pairs signed rank test for paired comparisons. *$P < 0.05$, **$P < 0.01$, ****$P < 0.0001$; NS, not significant. HS, human AB serum control.

the age of 80 years (Fig. 1a). Given this nonlinear change in a correlate of protection, we performed selected subsequent analyses with age both as a continuous variable and as a categorical variable. When individuals aged 80 years or more were tested between 3 and 12 weeks after their first dose, around half showed no evidence of neutralization (Extended Data Fig. 2). Geometric mean neutralization titre (GMT) was lower in participants aged 80 years or more than in younger individuals (48.2 (95% confidence interval (CI) 34.6–67.1) versus 104.1 (95% CI 69.7–155.2), $P = 0.004$; Extended Data Table 1, Fig. 1b). GMT showed evidence of an inverse association with age (Extended Data Fig. 2). The GMT following the second dose was significantly higher in individuals for whom there had been a 12-week interval between doses compared with a 3-week interval between doses (Extended Data Fig. 2). A clinically accredited assay for N antibodies[9] showed evidence that five individuals in each group had previously been infected with SARS-CoV-2 (Extended Data Table 1), and we adjusted for this in multivariable analyses (Extended Data Tables 2, 3). Neutralizing titres for sera from vaccinated individuals were higher after the second dose than after the first dose, regardless of age (Fig. 1b). In participants who had suboptimal or no neutralization after dose 1, and who subsequently received the second dose within the study period (Fig. 1c), all but two elderly participants responded with an increase in neutralization activity (Extended Data Table 1, Fig. 1b).

Given our observation that participants aged 80 years or more had lower neutralization responses following the first dose than younger individuals, we hypothesized that this could lead to sub-protective neutralizing responses against the B.1.1.7, B.1.351 and P.1 VOCs, which were first identified in the UK, South Africa and Brazil, respectively

(Extended Data Fig. 2). We therefore examined serum neutralization by age group against PVs bearing the wild-type spike protein or spike proteins from the three VOCs (Fig. 1d, e). There was a clear reduction in neutralizing titres against VOCs (Fig. 1d), and titres were lower for individuals over 80 years old than for younger individuals. The proportions of individuals with detectable neutralization showed a similar pattern (Fig. 1e, Extended Data Tables 2, 3). Following the second dose, although there were differences in GMT for the VOCs between the age groups (Fig. 1f), nearly all participants across age groups had detectable neutralization responses across the VOCs tested (Fig. 1g).

## B cell responses to mRNA vaccination

We measured binding antibody responses to the full-length wild-type (Wuhan-1) spike protein[9]. Levels of IgG and all IgG subclasses against spike protein increased between vaccine doses (Fig. 2a), and were similar after the second dose to those observed following natural infection. Like the neutralization titres, levels of IgG against spike declined with age (Fig. 2b, Extended Data Fig. 3). IgG and its subclasses correlated with serum neutralization (Fig. 2c, Extended Data Fig. 3). The concentrations of total and subclass anti-spike IgGs were significantly lower in participants aged 80 or older than in the younger group (Fig. 2d). IgA responses also increased between the two doses and correlated with neutralization after dose 1 (Extended Data Fig. 3). In addition, phenotyping of peripheral blood mononuclear cells (PBMCs) by flow cytometry showed that neutralization in the over-80 age group was associated with a higher

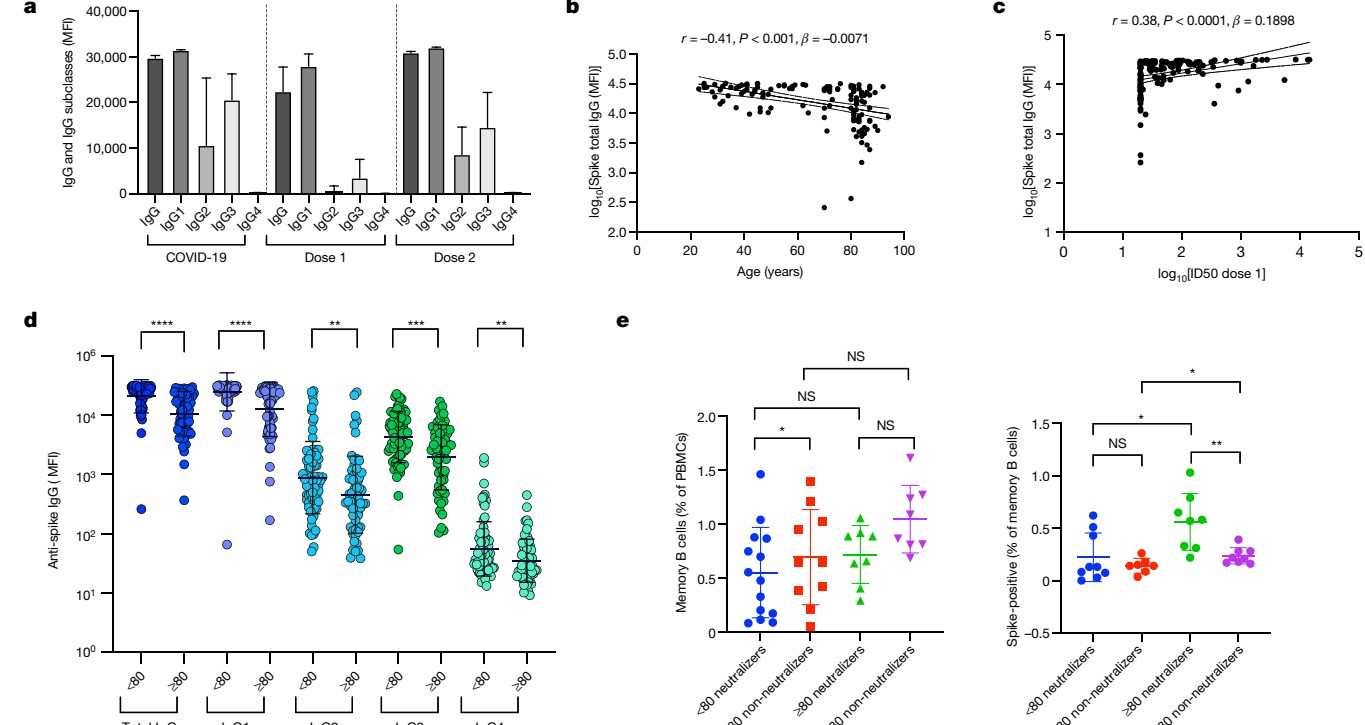

**Fig. 2 | SARS-CoV-2 spike-binding antibody responses and SARS-CoV-2 spike-specific memory B cells in blood following vaccination with BNT162b2. a**, Total anti-spike IgG and subclasses after first and second doses of vaccine and in individuals with prior COVID-19. MFI, mean fluorescence intensity. **b**, Pearson's correlation (*r*) between anti-spike IgG binding antibody responses after first dose and age (*n* = 134). **c**, Pearson's correlation between anti-spike IgG (*n* = 134) binding antibody responses and neutralization by sera against SARS-CoV-2 in a spike lentiviral pseudotyping assay expressing wild-type spike (D614G). **d**, Anti-spike IgG subclass responses to first dose

vaccine stratified by age (<80 and ≥80 years). **e**, CD19⁺ memory B cells (left, as percentage of PBMCs) and SARS-CoV-2 spike-specific CD19⁺IgG⁺IgM⁻ memory B cells (right, as percentage of all memory B cells) from FACS-sorted PBMCs. *n* = 16 for ≥80 years, *n* = 16 for <80 years; stratified by neutralizing response after first dose, *n* = 8 in each category. MFI – mean fluorescence intensity. Mann–Whitney test was used for unpaired comparisons. *$P$ < 0.05, **$P$ < 0.01, ***$P$ < 0.001, ****$P$ < 0.0001; NS, not significant. Scatter plots show linear correlation line bounded by 95% CI; $\beta$, slope/regression coefficient. Error bars, s.d.

proportion of spike-specific IgG⁺IgM⁻CD19⁺ memory B cells (Fig. 3e). Notably, the proportion of these cells did not differentiate neutralizers from non-neutralizers in the under-80 group (Fig. 3e, Extended Data Fig. 4).

We performed B cell repertoire sequencing on bulk PBMCs to assess isotype and variable gene usage, somatic hypermutation and diversity of the repertoire between the two age groups and in relation to neutralization. There were no differences in isotype proportions between the two age groups (Extended Data Fig. 5), or by neutralization (Fig. 3a). We found an increase in usage of the immunoglobulin heavy variable 4 (IGHV4) family in the older age group, with an increased proportion of IGHV4.34, IGHV4.39, IGHV4.59 and IGHV4.61, whereas in the younger age group there was an increase in usage of the IGHV1 family, with increases in IGHV1.18 and IGHV1.69D (Fig. 3b). We did not find any significant differences in V gene usage associated with neutralization (Extended Data Fig. 5).

Differences in somatic hypermutation could affect neutralization through antibody affinity maturation. We found that participants aged 80 years or more had a lower level of somatic hypermutation in class-switched B cell receptors (BCRs) than the younger group, and that the difference was driven by the IgA1/2 isotype (Fig. 3c). We also did not find any relationship between measures of diversity and neutralization potency or age group (Fig. 3d, Extended Data Fig. 5). We next examined the B cell repertoire for public clones known to be associated with SARS-CoV-2 neutralization. We explored the convergence between BCR clones in our study and the CoV-AbDab database[12] and found that participants under 80 years of age had a higher frequency of convergent clones, in keeping with increased neutralization, when compared with the older group (Fig. 3e).

## T cell responses to mRNA vaccination

Although it is increasingly recognized that neutralizing antibodies dominate protection against initial infection[4,13], T cells might limit disease progression[5] when neutralizing antibody titres are low[14]. We therefore determined the T cell response to SARS-CoV-2 spike protein in vaccinated individuals by stimulating PBMCs with overlapping peptide pools to the wild-type SARS-CoV-2 spike, using an interferon-γ (IFNγ) and interleukin-2 (IL-2) FluoroSpot assay to count spike-specific T cells. When we plotted IFNγ-spike specific T cell responses against age as a continuous variable, there was a negative correlation with a drop-off at around 80 years (Fig. 4a). A similar effect, albeit less pronounced, was seen for IL-2 (Fig. 4b). However, there did not appear to be a relationship between cytokine production by PBMCs and neutralization titre after the first dose (Extended Data Fig. 6).

Following the first dose of vaccine, the frequency of IFNγ-secreting T cells against a CEF+ peptide pool that included cytomegalovirus (CMV)-, Epstein-Barr virus (EBV)- and influenza-specific peptides did not differ by age category and was similar to healthy SARS-CoV-2 unexposed controls (Extended Data Fig. 6). This indicates that differences in observed responses were likely to be vaccine-specific rather than resulting from generalized suboptimal T cell responses or immune paresis. However, IFNγ spike-specific T cell responses were significantly larger in immunized individuals below 80 years of age than in an unexposed population of the same age (Fig. 4c). However, in participants aged 80 years or more, the IFNγ spike-specific T cell response following the first dose did not differ from that of unexposed controls (Fig. 4c). By contrast, spike-specific IL-2 T cell frequencies were

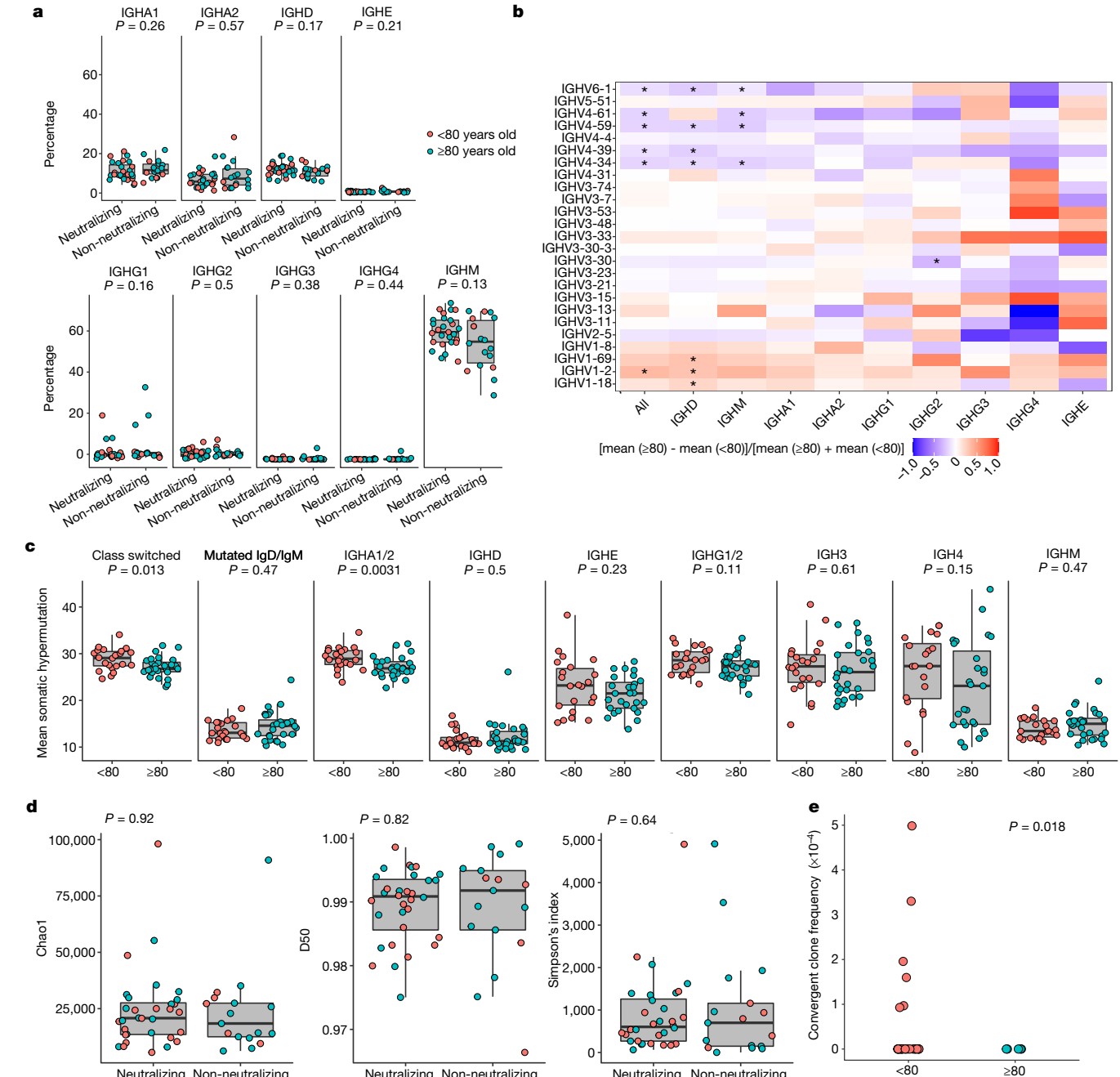

**Fig. 3 | B cell repertoire following vaccination with first dose of BNT162b2.**
**a**, Isotype usage according to unique VDJ sequence in participants <80 (*n* = 22) or ≥80 years old (*n* = 28) and association with neutralization of spike pseudotyped virus. Neutralization cut-off for 50% neutralization was set at 20. Mann–Whitney *U*-test. **b**, Heat map showing differences in V gene usage between the <80 and ≥80 groups. Mann–Whitney *U*-test with Benjamini–Hochberg false discovery rate (FDR) correction; *$P$ < 0.1. **c**, Mean somatic hypermutation for participants <80 or ≥80 years old, grouped according to isotype class. Mann–Whitney *U*-test. **d**, Diversity indices for neutralizing and non-neutralizing groups. The inverse is depicted for Simpson's index. *t*-test. **e**, BCR comparison of patients in the two age groups for the first 50 days after vaccination (<80, *n* = 27; ≥80, *n* = 5) with public clones known to be associated with SARS-CoV-2 using the CoV-AbDab database[12]. Clones from participants and the database were co-clustered based on matching IGHV and IGHJ segments, matching CDR-H3 region length and 85% CDR-H3 sequence amino acid homology. One-sided *t*-test. For boxplots: centre line, median; box, 25th–75th percentile; whiskers, 1.5× IQR.

significantly higher in both vaccinated groups than in unexposed controls (Fig. 4d). Notably, although spike-specific IFNγ and IL-2 responses in PBMCs after the first dose of vaccine were similar to those found after natural infection (Extended Data Fig. 6), the second dose did not appear to increase these responses, either overall (Extended Data Fig. 6) or within age categories (Fig. 4e, f). Following depletion of CD4 or CD8 T cells, the majority of IFNγ and IL-2 production was from CD4+ T cells in vaccinated individuals (Fig. 4g, h). Those aged 80 or more had markedly lower spike-specific IL-2 CD4+ T cell responses than their younger counterparts (Fig. 4g).

CMV serostatus has been associated with poorer responses to vaccination and infections[15,16]. The rate of CMV IgG positivity was higher in the older age group (Extended Data Fig. 6); unexpectedly, though, CMV-positive individuals in this group had significantly higher IFNγ, but not IL-2, responses to SARS-CoV-2 spike peptides than CMV-negative individuals in the same age group (Extended Data Fig. 6).

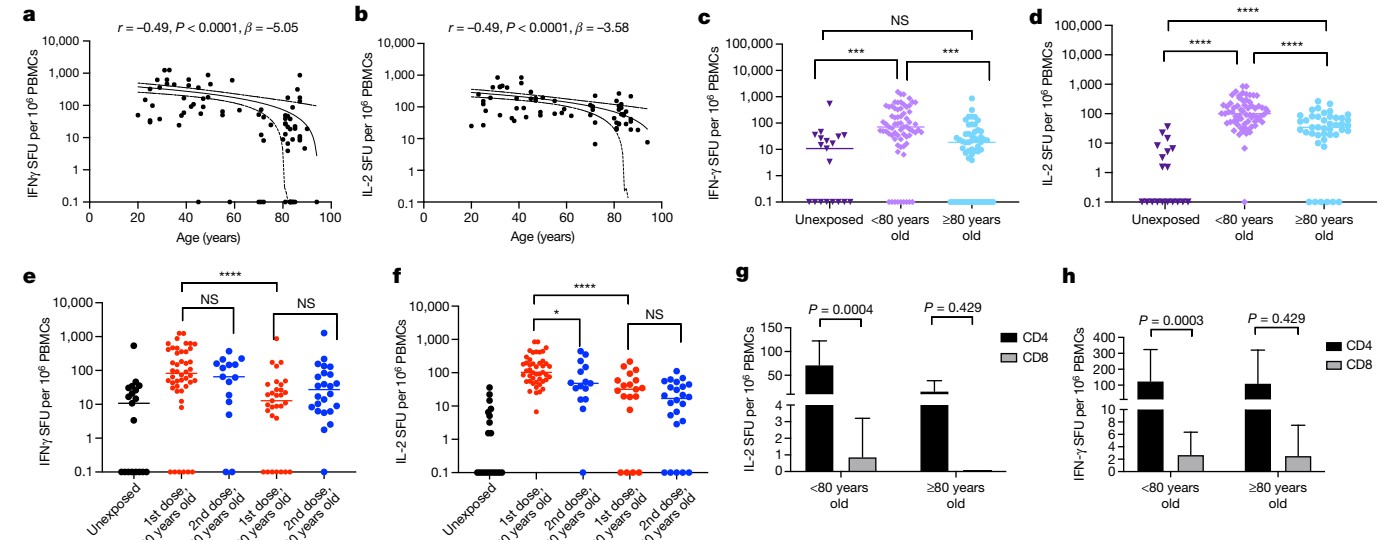

**Fig. 4 | T cell responses to BNT162b2 vaccine after the first and second doses. a**, **b**, FluoroSpot analysis by age for IFNγ (**a**) and IL-2 (**b**) T cell responses specific to SARS-CoV-2 spike protein peptide pool following PBMC stimulation. SFU, spot-forming units. Scatter plots show linear correlation line bounded by 95% CI; $\beta$, slope/regression coefficient. **c**, **d**, FluoroSpot analysis for IFNγ (**c**) and IL-2 (**d**) T cell responses specific to SARS-CoV-2 spike protein peptide pool following stimulation of unexposed PBMCs (stored PBMCs from 2014–2016, *n* = 20) and PBMCs from vaccinated individuals (<80 IFNγ, *n* = 46; <80 IL-2, *n* = 44; ≥80 IFNγ, *n* = 35; ≥80 IL-2, *n* = 27) three weeks or more after the first dose of BNT162b2. **e**, **f**, FluoroSpot analysis for IFNγ (**e**) and IL-2 (**f**) T cell responses

specific to SARS-CoV-2 spike protein peptide pool following stimulation of unexposed PBMCs (*n* = 20) and PBMCs from vaccinated individuals three weeks after the first or second dose (first dose: <80 IFNγ, *n* = 46; <80 IL-2, *n* = 45; ≥80 IFNγ, *n* = 31; ≥80 IL-2, *n* = 19; second dose: <80 IFNγ, *n* = 15; <80 IL-2, *n* = 15; ≥80 IFNγ, *n* = 24; ≥80 IL-2, *n* = 24). **g**, **h**, FluoroSpot analysis for IL-2 (**g**) and IFNγ (**h**) CD4 and CD8 T cell responses specific to SARS-CoV-2 spike protein peptide pool following stimulation after column-based PBMC separation. Mann–Whitney test was used for unpaired comparisons and Wilcoxon matched-pairs signed rank test for paired comparisons. *P < 0.05, ***P < 0.001, ****P < 0.0001; NS, not significant. Error bars, s.d.

## Autoantibodies and inflammatory molecules

Finally, we investigated the possibility of interactions between senescence and mRNA vaccine responses. Autoantibodies and inflammatory cytokines or chemokines are associated with immune senescence[17]. We first measured a panel of autoantibodies in the sera of 101 participants following the first dose of the BNT162b2 vaccine. Eight participants had autoantibodies against myeloperoxidase (anti-MPO), two against fibrillarin and one against cardiolipin (Extended Data Fig. 7). As expected, all but one of the participants with anti-MPO autoantibodies were over the age of 80 years (Extended Data Fig. 7). There was a trend towards reduced anti-spike IgG levels and serum neutralization against the wild-type and B.1.17 spike proteins in participants with autoantibodies, although this did not reach statistical significance (probably owing to the small sample size; Extended Data Fig. 7). Next, we explored the association between serum cytokines or chemokines and neutralization of SARS-CoV-2 PV, as well as their association with age. PIDF, a known senescence-associated secretory phenotype (SASP) molecule, was the only molecule that was enriched in sera from participants aged over 80 years, and there was no association between any of these molecules and the ability of sera to neutralize SARS-CoV-2 PVs (Extended Data Fig. 7).

## Discussion

Neutralizing antibodies are a likely correlate of protection against SARS-CoV-2 infection, as suggested by vaccine efficacy studies, pre-clinical studies in mice and non-human primates, and data from the early use of convalescent plasma in elderly patients[4,5,10,13,14,18,23]. There is a lack of data on neutralizing antibody immune responses following mRNA vaccination in the elderly, and no data, to our knowledge, on variants of concern in this group. In a clinical study that specifically looked at older adults vaccinated with BNT162b2, the GMT after the first dose was 12 in a set of 12 subjects between ages of 65 and 85 years,

rising to 149 seven days after the second dose[1]. Furthermore, in a study of the Moderna 1273 mRNA vaccine in individuals above 55 years of age, neutralization was detectable only after the second dose, whereas binding antibodies were detectable after both doses[19]. In a randomized phase I study on BNT162b1 in younger (18–55 years) and older adults (65–85 years), virus neutralization was lower in the older age group 22 days after the first dose[20]. These data reflect the finding that responses to the ChAdOx1 nCov-19 (AZD-1222) vaccine were lower in older than in younger mice, and the difference was overcome by booster dosing[21].

Here, in a cohort of 140 individuals, we have shown not only an inverse relationship between age and neutralizing responses following the first dose of BNT162b2, but also a more precipitous decline around the age of 80 years. Individuals aged 80 or more were prioritized for vaccination in the UK and elsewhere, as they represented the group at greatest risk of severe COVID-19[22]. We found that around half of those above the age of 80 have a suboptimal neutralizing antibody response after the first dose of BNT162b2, accompanied by lower T cell responses compared to younger individuals. Individuals over 80 years of age differed from the younger group in four main respects that could explain poorer neutralization of SARS-CoV-2. First, serum IgG levels were lower, accompanied by a lower proportion of peripheral spike-specific IgG+IgM−CD19+ memory B cells. Second, the elderly displayed lower somatic hypermutation in the *BCR* gene. Third, the elderly had lower enrichment for public BCR clonotypes that are associated with neutralization. And fourth, the older group displayed a marked reduction in IL-2-producing spike-reactive CD4+ T cells. Therefore, possible explanations for their poorer neutralizing responses include lower concentrations of antibodies (quantity) and/or lower-affinity antibodies (quality) resulting from B cell selection, reduced CD4+ T cell help, or a combination of both. These data parallel those in aged mice, where ChAdOx1 nCov-19 (AZD-1222) vaccine responses were reported to be lower than in younger mice, and this was overcome by booster dosing[21].

Critically, we show that elderly individuals are likely to be at greater risk from VOCs, as a greater proportion of individuals in the over-80 age

group showed no neutralizing activity to P.1 and B.1.1.7 after the first dose. Reassuringly, we observed neutralizing responses across all age groups after the second dose, although further work is needed to understand the effect of age on the durability of immune responses following vaccination.

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

**The CITIID-NIHR BioResource COVID-19 Collaboration**

**Principal Investigators**
Stephen Baker[2,3], Gordon Dougan[2,3], Christoph Hess[2,3,17,18], Nathalie Kingston[11], Paul J. Lehner[2,3], Paul A. Lyons[2,3], Nicholas J. Matheson[2,3], Willem H. Owehand[11], Caroline Saunders[10], Charlotte Summers[3,15,16,19], James E. D. Thaventhiran[2,3,13], Mark Toshner[3,15,16], Michael P. Weekes[2], Patrick Maxwell[11,19] & Ashley Shaw[19]

**CRF and Volunteer Research Nurses**
Ashlea Bucke[10], Jo Calder[10], Laura Canna[10], Jason Domingo[10], Anne Elmer[10], Stewart Fuller[10], Julie Harris[22], Sarah Hewitt[10], Jane Kennet[10], Sherly Jose[10], Jenny Kourampa[10], Anne Meadows[10], Criona O'Brien[10], Jane Price[10], Cherry Publico[10], Rebecca Rastall[10], Carla Ribeiro[10], Jane Rowlands[10], Valentina Ruffolo[10] & Hugo Tordesillas[10]

**Sample Logistics**
Ben Bullman[2], Benjamin J. Dunmore[3], Stuart Fawke[21], Stefan Gräf[3,11], Josh Hodgson[3], Christopher Huang[3], Kelvin Hunter[2,3], Emma Jones[20], Ekaterina Legchenko[3], Cecilia Matara[3], Jennifer Martin[3], Federica Mescia[2,3], Ciara O'Donnell[3], Linda Pointon[3], Nicole Pond[2,3], Joy Shih[3], Rachel Sutcliffe[3], Tobias Tilly[3], Carmen Treacy[3], Zhen Tong[3], Jennifer Wood[3] & Marta Wylot[27]

**Sample Processing and Data Acquisition**
Laura Bergamaschi[2,3], Ariana Betancourt[2,3], Georgie Bower[2,3], Chiara Cossetti[2,3], Aloka De Sa[3], Madeline Epping[2,3], Stuart Fawke[11], Nick Gleadall[11], Richard Grenfell[22], Andrew Hinch[2,3], Oisin Huhn[23], Sarah Jackson[3], Isobel Jarvis[3], Ben Krishna[3], Daniel Lewis[3], Joe Marsden[3], Francesca Nice[3], Georgina Okecha[3], Ommar Omarjee[3], Marianne Perera[3], Martin Potts[3], Nathan Richoz[3], Veronika Romashova[2,3], Natalia Savinykh Yarkoni[3], Rahul Sharma[3], Luca Stefanucci[11], Jonathan Stephens[11], Mateusz Strezlecki[22] & Lori Turner[2,3]

**Clinical Data Collection**
Eckart M. D. D. De Bie[3], Katherine Bunclark[3], Masa Josipovic[3], Michael Mackay[3], Federica Mescia[2,3], Alice Michael[16], Sabrina Rossi[26], Mayurun Selvan[3], Sarah Spencer[14] & Cissy Yong[26]

**Royal Papworth Hospital ICU**
Ali Ansaripour[16], Alice Michael[16], Lucy Mwaura[16], Caroline Patterson[16] & Gary Polwarth[16]

**Addenbrooke's Hospital ICU**
Petra Polgarova[19] & Giovanni di Stefano[19]

**Cambridge and Peterborough Foundation Trust**
Codie Fahey[25] & Rachel Michel[25]

**ANPC and Centre for Molecular Medicine and Innovative Therapeutics**
Sze-How Bong[12], Jerome D. Coudert[24] & Elaine Holmes[28]

**NIHR BioResource[4]**
John Allison[4,11], Helen Butcher[4,11], Daniela Caputo[5,11], Debbie Clapham-Riley[5,11], Eleanor Dewhurst[5,11], Anita Furlong[5,11], Barbara Graves[5,11], Jennifer Gray[5,11], Tasmin Ivers[5,11], Mary Kasanicki[5], Emma Le Gresley[5,11], Rachel Linger[5,11], Sarah Meloy[5,11], Francesca Muldoon[5,11], Nigel Ovington[5,11], Sofia Papadia[5,11], Isabel Phelan[5,11], Hannah Stark[5,11], Kathleen E. Stirrups[4], Paul Townsend[4], Neil Walker[4] & Jennifer Webster[5,11]

[10]Cambridge Clinical Research Centre, NIHR Clinical Research Facility, Cambridge University Hospitals NHS Foundation Trust, Addenbrooke's Hospital, Cambridge, UK. [11]University of Cambridge, Cambridge Biomedical Campus, Cambridge, UK. [12]Australian National Phenome Centre, Murdoch University, Murdoch, Western Australia, Australia. [13]MRC Toxicology Unit, School of Biological Sciences, University of Cambridge, Cambridge, UK. [14]R&D Department, Hycult Biotech, Uden, The Netherlands. [15]Heart and Lung Research Institute, Cambridge Biomedical Campus, Cambridge, UK. [16]Royal Papworth Hospital NHS Foundation Trust, Cambridge Biomedical Campus, Cambridge, UK. [17]Department of Biomedicine, University and University Hospital Basel, Basel, Switzerland. [18]Botnar Research Centre for Child Health (BRCCH), University Basel & ETH Zurich, Basel, Switzerland. [19]Addenbrooke's Hospital, Cambridge, UK. [20]Department of Veterinary Medicine, Cambridge, UK. [21]Cambridge Institute for Medical Research, Cambridge Biomedical Campus, Cambridge, UK. [22]Cancer Research UK, Cambridge Institute, University of Cambridge, Cambridge, UK. [23]Department of Obstetrics & Gynaecology, The Rosie Maternity Hospital, Cambridge, UK. [24]Centre for Molecular Medicine and Innovative Therapeutics, Health Futures Institute, Murdoch University, Perth, Western Australia, Australia. [25]Cambridge and Peterborough Foundation Trust, Fulbourn Hospital, Cambridge, UK. [26]Department of Surgery, Addenbrooke's Hospital, Cambridge, UK. [27]Department of Biochemistry, University of Cambridge, Cambridge, UK. [28]Centre of Computational and Systems Medicine, Health Futures Institute, Murdoch University, Perth, Western Australia, Australia.

## Methods

### Study design

Community participants or healthcare workers who received their first dose of the BNT162b2 vaccine between 14 December 2020 and 10 February 2021 were consecutively recruited at Addenbrooke's Hospital into the COVID-19 cohort of the NIHR Bioresource. Participants were followed up for up to 3 weeks after receiving their second dose of the BNT162b2 vaccine. They provided blood samples 3 to 12 weeks after their first dose and again 3 weeks after the second dose of the vaccine. Consecutive participants were eligible without exclusion. The exposure of interest was age, categorized into two exposure levels (<80 and ≥80 years). The outcome of interest was inadequate vaccine-elicited serum antibody neutralization activity at least 3 weeks after the first dose. This was measured as the dilution of serum required to inhibit infection by 50% (ID50) in an in vitro neutralization assay. An ID50 of 20 or below was deemed inadequate neutralization. Binding antibody responses to the spike, receptor-binding domain (RBD) and nucleocapsid were measured by multiplex particle-based flow cytometry and spike-specific T cell responses were measured by IFNγ and IL-2 FluoroSpot assays. Measurement of serum autoantibodies and characterization of the B cell receptor (BCR) repertoire following the first vaccine dose were exploratory outcomes.

We assumed a risk ratio of non-neutralization in the ≥80 years group compared with the <80 years group of 5. Using an alpha of 0.05 and power of 90% required a sample size of 50 with a 1:1 ratio in each group.

### Ethical approval

The study was approved by the East of England – Cambridge Central Research Ethics Committee (17/EE/0025). PBMCs from unexposed volunteers previously recruited by the NIHR BioResource Centre Cambridge through the ARIA study (2014–2016) were used with ethical approval from the Cambridge Human Biology Research Ethics Committee (HBREC.2014.07) and currently North of Scotland Research Ethics Committee 1 (NS/17/0110).

### Statistical analyses

Descriptive analyses of demographic and clinical data are presented as median and IQR when continuous and as frequency and proportion (%) when categorical. Differences between continuous and categorical data were tested using Wilcoxon rank sum and Chi-square tests, respectively. Logistic regression was used to model the association between age group and neutralization by vaccine-elicited antibodies after the first dose of the BNT162b2 vaccine. The effects of sex and time interval from vaccination to sampling as confounders were adjusted for. Linear regression was also used to explore the association between age as a continuous variable and log-transformed ID50, binding antibody levels, antibody subclass levels and T cell response after dose 1 and dose 2 of the BNT162b2 vaccine. Bonferroni adjustment was made for multiple comparisons in the linear correlation analyses between binding antibody levels, ID50, age and T cell responses. The Pearson's normally distributed correlation coefficient for linear data and Spearman's non-normally distributed correlation for nonlinear data were reported. Statistical analyses were done using Stata v13, Prism v9 and R (version 3.5.1).

### Generation of mutants and pseudotyped viruses

Wild-type pseudotyped virus (bearing mutation D614G), B.1.1.7 pseudotyped viruses (bearing mutations Δ69/70, Δ144, N501Y, A570D, D614G, P681H, T716I and S982A and D1118H), B.1.351 pseudotyped virus (bearing mutations L18F, D80A, D215G, Δ242-4, R246I, K417N, E484K, N501Y, D614G, A701V) and P.1 pseudotyped virus (bearing mutations L18F, T20N, P26S, D138Y, R190S, K417T, E484K, N501Y, D614G, H655Y, T1027I and V1176F) were generated. In brief, amino acid substitutions were introduced into the D614G pCDNA_SARS-CoV-2_S plasmid as previously described[23] using the QuikChange Lightening Site-Directed Mutagenesis kit, according to the manufacturer's instructions (Agilent Technologies). Sequences were verified by Sanger sequencing. The pseudoviruses were generated in a triple plasmid transfection system whereby the spike-expressing plasmid along with a lentiviral packaging vector (p8.9) and luciferase expression vector (psCSFLW) were transfected into 293T cells (a gift from Greg Towers; tested for mycoplasma) with Fugene HD transfection reagent (Promega). The viruses were harvested after 48 h and stored at −80 °C. TCID50 was determined by titration of the viruses on 293T cells expressing ACE-2 and TMPRSS2[24].

### Pseudotyped virus neutralization assays

Spike pseudotype assays have been shown to have similar characteristics as neutralization testing using fully infectious wild-type SARS-CoV-2[11]. Virus neutralization assays were performed on 293T cells transiently transfected with ACE2 and TMPRSS2 using SARS-CoV-2 spike pseudotyped virus expressing luciferase[24]. Pseudotyped virus was incubated with serial dilutions of heat-inactivated human serum samples or sera from vaccinated individuals in duplicate for 1 h at 37 °C. Virus and cell-only controls were also included. Then, freshly trypsinized 293T ACE2/TMPRSS2-expressing cells were added to each well. Following 48 h incubation with 5% $CO_2$ at 37 °C, luminescence was measured using the Steady-Glo Luciferase assay system (Promega). Neutralization was calculated relative to virus-only controls. Dilution curves were presented as a mean neutralization with s.e.m. ID50 values were calculated in GraphPad Prism. The limit of detection for 50% neutralization was set at an ID50 of 20. The ID50 within groups were summarized as a geometric mean titre (GMT) and statistical comparison between groups were made with Mann–Whitney or Wilxocon ranked sign tests.

### Live virus serum neutralization assays

A549-ACE2-TMPRSS2 cells were seeded at a cell density of $2.4 \times 10^4$ per well in a 96-well plate 24 h before inoculation. Serum was titrated starting at a final dilution of 1:50 with live B.1 virus PHE2 (EPI_ISL_407073) isolate being added at a multiplicity of infection (MOI) of 0.01. The mixture was then incubated for 1 h before being added to the cells. Seventy-two hours after infection, the plates were fixed with 8% formaldehyde and then stained with Coomassie blue for 30 min. The plates were washed and dried overnight before using a Celigo Imaging Cytometer (Nexcelom) to measure the staining intensity. Percentage cell survival was determined by comparing the intensity of the staining to an uninfected well. A nonlinear sigmoidal 4PL model (Graphpad Prism 9) was used to determine the $IC_{50}$ for each serum. The correlation between log-transformed ID50 obtained from the pseudotyped virus and live virus systems were explored using linear regression. Pearson's correlation coefficient was determined.

### SARS-CoV-2 serology by multiplex particle-based flow cytometry (Luminex)

Recombinant SARS-CoV-2 nucleocapsid, spike and RBD were covalently coupled to distinct carboxylated bead sets (Luminex) to form a 3-plex and analysed as previously described[25]. Specific binding was reported as mean fluorescence intensities (MFI).

### CMV serology

HCMV IgG levels were determined using an IgG enzyme-linked immunosorbent assay (ELISA), HCMV Captia (Trinity Biotech) according to the manufacturer's instructions, on plasma derived from clotted blood samples.

### Serum autoantibodies

Serum was screened for the presence of autoantibodies using the ProtoPlex autoimmune panel (Life Technologies) according to the manufacturer's instructions. In brief, 2.5 μl serum was incubated with Luminex MagPlex magnetic microspheres in a multiplex format conjugated to

19 full-length human autoantigens (cardiolipin, CENP B, H2a(F2A2) and H4 (F2A1), Jo-1, La/SS-B, Mi-2b, myeloperoxidase, proteinase-3, pyruvate dehydrogenase, RNP complex, Ro52/SS-A, Scl-34, Scl-70, Smith antigen, thyroglobulin, thyroid peroxidase, transglutaminase, U1-snRNP 68, and whole histone) along with bovine serum albumin (BSA). Detection was undertaken using goat-anti-human IgG-RPE in a 96-well flat-bottomed plate and the plate was read in a Luminex xMAP 200 system. Raw fluorescence intensities (FI) were further processed in R (version 3.5.1) Non-specific BSA-bound FI was subtracted from background-corrected total FI for each antigen before $log_2$ transformation and thresholding. Outlier values (Q3 + 1.5 × IQR) in each distribution were defined as positive.

## Serum chemokine and cytokine analysis

Serum proteins were quantified using a validated electrochemiluminescent sandwich assay quantification kit (Mesoscale Discovery VPlex) according to the manufacturer's instructions. In brief, both sera and standard calibration controls were incubated with SULFO-tagged antibodies targeting IFNγ, IL-10, IL-12p70, IL-13, IL-1β, IL-2, IL-4, IL-6, IL-8, TNFα, GC-CSF, IL-1α, IL-12, IL-15, IL-16, IL-17A, IL-5, IL-7, TNFβ, VEGF, MCP1, MCP4, eotaxin, eotaxin3, IP10, MDC, MIP1α, MIP1β, TARC, IL-17B, IL-17C, IL-17D, IL-1RA, IL-3, IL-9, TSLP, VEGFA, VEGFC, VEGFD, VEGFR1/FLT1, PIGF, TIE2, FGF, ICAM1, VCAM1, SAA and CRP and read using an MSD MESO S600 instrument. Concentrations were calculated by comparison with an internal standard calibration curve fitted to a four-parameter logistic model. Values below (19%) or above (0%) the reference range were imputed at the lower/upper limit of detection, respectively. Association of each cytokine level with SARS-CoV-2 neutralizing antibody titre, neutralization status (1/0) and age was undertaken using Kendall's tau and Wilcoxon tests with FDR <5% considered significant.

## B cell receptor repertoire library preparation

PBMCs were lysed and RNA extracted using Qiagen AllPrep DNA/RNA mini kits and Allprep DNA/RNA Micro kits according to the manufacturer's protocol. The RNA was quantified using a Qubit. B cell receptor repertoire libraries were generated for 52 COVID-19 vaccinated individuals (58 samples) as follows: 200 ng total RNA from PAXgenes (14 µl volume) was combined with 1 µl 10 mM dNTP and 10 µM reverse primer mix (2 µl) and incubated for 5 min at 70 °C. The mixture was immediately placed on ice for 1 min and then subsequently combined with 1 µl DTT (0.1 M), 1 µl SuperScriptIV (Thermo Fisher Scientific), 4 µl SSIV Buffer (Thermo Fisher Scientific) and 1 µl RNase inhibitor. The solution was incubated at 50 °C for 60 min followed by 15 min inactivation at 70 °C. cDNA was cleaned with AMPure XP beads and PCR-amplified with a 5′ V-gene multiplex primer mix and 3′ universal reverse primer using the KAPA protocol and the following thermal cycling conditions: 1 cycle (95 °C, 5 min); 5 cycles (98 °C, 20 s; 72 °C, 30 s); 5 cycles (98 °C, 15 s; 65 °C, 30 s; 72 °C, 30 s); 19 cycles (98 °C, 15 s; 60 °C, 30 s; 72 °C, 30 s); 1 step (72 °C, 5 min). Sequencing libraries were prepared using Illumina protocols and sequenced using 300-bp paired-end sequencing on a MiSeq machine.

## Sequence analysis

Raw reads were filtered for base quality using a median Phred score of ≥ 32 (http://sourceforge.net/projects/quasr/). Forward and reverse reads were merged where a minimum 20-bp identical overlapping region was present. Sequences were retained where more than 80% base sequence similarity was present between all sequences with the same barcode. The constant-region allele with highest sequence similarity was identified by 10-mer matching to the reference constant-region genes from the IMGT database. Sequences without complete reading frames and non-immunoglobulin sequences were removed and only reads with significant similarity to reference IGHV and J genes from the IMGT database using BLAST were retained. Immunoglobulin gene use and sequence annotation were performed in IMGT V-QUEST, and repertoire differences were analysed by custom scripts in Python.

## Public BCR analysis

Convergent clones were annotated with the same IGHV and IGHJ segments, had the same CDR-H3 region length and were clustered based on 85% CDR-H3 sequence amino acid homology. A cluster was considered convergent with the CoV-AbDab database if it contained sequences from post-vaccinated individuals and from the database.

## Flow cytometry

The following antibodies or staining reagents were purchased from BioLegend: CD19 (SJ25C, 363028), CD3 (OKT3, 317328), CD11C (3.9, 301608), CD25 (M-A251, 356126), CD14 (M5E2,301836), and IgM (IgG1-k, 314524). CCR7 (150503, 561143) and IgG (G18-145, 561297) were obtained from BD Bioscience, CD45RA (T6D11, 130-113-359) from Miltyeni Biotech, and CD8A (SK1, 48-0087-42) from eBiosciences. The LIVE/DEAD Fixable Aqua Dead Cell Stain Kit was obtained from Invitrogen. Biotinylated spike protein expressed and purified as previously described[26] was conjugated to streptavidin R-phycoerythrin (PJRS25-1) or streptavidin APC obtained from Agilent Technologies. PBMCs were isolated from study participants and stored in liquid nitrogen. Aliquots containing $10^7$ cells were thawed and stained in PBS containing 2 mM EDTA at 4 °C with the above antibody panel and then transferred to 0.04% BSA in PBS. Events were acquired on a FACSAria Fusion (BD Biosciences). Analyses were carried out in FlowJo version 10.7.1.

## IFNγ and IL-2 FluoroSpot T cell assays

PBMCs were isolated from the heparinized blood samples using Histopaque-1077 (Sigma-Aldrich) and SepMate-50 tubes (StemCell Technologies). Frozen PBMCs were rapidly thawed and diluted into 10 ml TexMACS medium (Miltenyi Biotech), centrifuged and resuspended in 10 ml fresh medium with 10 U/ml DNase (Benzonase, Merck-Millipore via Sigma-Aldrich). PBMCs were then incubated at 37 °C for 1 h, followed by centrifugation and resuspension in fresh medium supplemented with 5% human AB serum (Sigma Aldrich) before being counted. PBMCs were stained with 2 µl LIVE/DEAD Fixable Far Red Dead Cell Stain Kit (Thermo Fisher Scientific) and live PBMCs were enumerated on the BD Accuri C6 flow cytometer.

## Overlapping spike SARS-CoV-2 peptide stimulation

A peptide pool was generated using the following: 1. PepTivator SARS-CoV-2 Prot_S containing the sequence domains (amino acids) 304–338, 421–475, 492–519, 683–707, 741–770, 785–802, and 885–1,273 and the N-terminal S1 domain of the surface glycoprotein (S) of SARS-CoV-2 (GenBank MN908947.3, Protein QHD43416.1). 2. The PepTivator SARS-CoV-2 Prot_S1 containing amino acids 1–692. The peptides used are 15 amino acids with 11-amino acid overlaps.

We incubated 1.0 to 2.5 × $10^5$ PBMCs from vaccinated individuals in pre-coated FluoroSpot[FLEX] plates (anti-IFNγ and anti-IL-2 capture antibodies, Mabtech) in duplicate with the spike peptide pool mix as described above (specific for Wuhan-1, QHD43416.1 spike SARS-CoV-2 protein; Miltenyi Biotech) or a mixture of peptides specific for cytomegalovirus, Epstein–Barr virus and influenza virus (CEF+, Miltenyi Biotech) (final peptide concentration as recommended by the manufacturer: 1 µg/ml/peptide) in addition to an unstimulated (medium only) and positive control mix (containing anti-CD3 (Mabtech AB) and *Staphylococcus* Enterotoxin B (SEB, Sigma Aldrich)) at 37 °C in a humidified $CO_2$ atmosphere for 42 h. The cells and medium were then decanted from the plate and the assay developed according to the manufacturer's instructions. Developed plates were read using an AID iSpot reader (Oxford Biosystems) and counted using AID EliSpot v7 software (Autoimmun Diagnostika). Peptide-specific frequencies were calculated by subtracting for background cytokine-specific spots (unstimulated control) and expressed as SFU per $10^6$ PBMCs. With the same peptide pool, we also stimulated PBMC that had been collected and biobanked between 2014

and 2016, representing a healthy population that had not been exposed to SARS-CoV-2, and PBMCs from donors who had been infected with SARS-CoV-2 (confirmed by RT–PCR) for comparison of T cell responses following natural infection.

## CD4 and CD8 depletion from PBMCs for subsequent FluoroSpot analysis

Peripheral blood mononuclear cells were depleted of either CD4+ or CD8+ T cells by magnetic-activated cell sorting (MACS) using anti-CD4 or anti-CD8 direct beads (Miltenyi Biotec), according to the manufacturer's instructions, and separated using an AutoMACS Pro (Miltenyi Biotec). The efficiency of depletion was determined by staining cells with a mix of CD3-FITC, CD4-PE, and CD8-PerCPCy5.5 antibodies (all BioLegend) and analysing by flow cytometry.

## Reporting summary

Further information on research design is available in the Nature Research Reporting Summary linked to this paper.

## Data availability

Sequence data have been deposited at the European Genome-Phenome Archive (https://ega-archive.org/) which is hosted by the EBI and the CRG under accession number EGAS00001005380. Data are available without restriction.

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

**Acknowledgements** We thank the Cambridge University Hospitals NHS Trust Occupational Health Department; the NIHR Cambridge Clinical Research Facility and staff at CUH; the Cambridge NIHR BRC Stratified Medicine Core Laboratory NGS Hub; the NIHR Cambridge BRC Phenotyping Hub; P. Mlcochova, S. A. Kemp, M. Potts, B. Krishna, M. Perera and G. Okecha; and J. Nathan, L. James and J. Briggs. R.K.G. is supported by a Wellcome Trust Senior Fellowship in Clinical Science (WT108082AIA). D.A.C. is supported by a Wellcome Trust Clinical PhD Research Fellowship. K.G.C.S. is the recipient of a Wellcome Investigator Award (200871/Z/16/Z). M.A.L. is supported by the Biotechnology and Biological Sciences Research Council (BBS/E/B/000C0427, BBS/E/B/000C0428). This research was supported by the National Institute for Health Research (NIHR) Cambridge Biomedical Research Centre, the Cambridge Clinical Trials Unit (CCTU), the NIHR BioResource and Addenbrooke's Charitable Trust, the Evelyn Trust (20/75), and the UKRI COVID Immunology Consortium. G.B.-M. and P.C.-A. were supported by UNAM-FESC-PIAPI Program Code PIAPI2009 and by a CONACyT 829997 Fellowship. The views expressed are those of the authors and not necessarily those of the NIHR or the Department of Health and Social Care. I.A.T.M.F. is funded by a Sub-Saharan African Network for TB/HIV Research Excellence (SANTHE, a DELTAS Africa Initiative (grant DEL-15–006)) Fellowship. We thank D. Corti for the VOC plasmids. P.K. is the recipient of a Jacquot Research Entry Scholarship from the Royal Australasian College of Physicians Foundation.

**Author contributions** Conceived study: R.K.G., D.A.C., I.A.T.M.F., E.M., R.D., M.W. Designed study and experiments: R.K.G., I.A.T.M.F., P.K., D.A.C., L.E.M., J.R.B., E. Thomson, M.W., D.C., E.L.G., M.A.L., L.C.-G., G.B.-M., R.P.D., A.E. Performed experiments: B.M., E.Y.L., L.B., A.A., P.C.-A., G.B.-M., P.A.L., C.D., D.A.C., E. Touizer, R.P.D., P.K., I.A.T.M.F., N.K., B.G., L.C.-G.. Interpreted data: R.K.G., D.A.C., B.M., R.P.D., E.M., R.D., I.A.T.M.F., M.W., E.Y.L., L.E.M., J.R.B., K.G.C.S.

**Competing interests** The authors declare no competing interests.

**Additional information**
**Correspondence and requests for materials** should be addressed to E.M., R.D., M.W. or R.K.G.

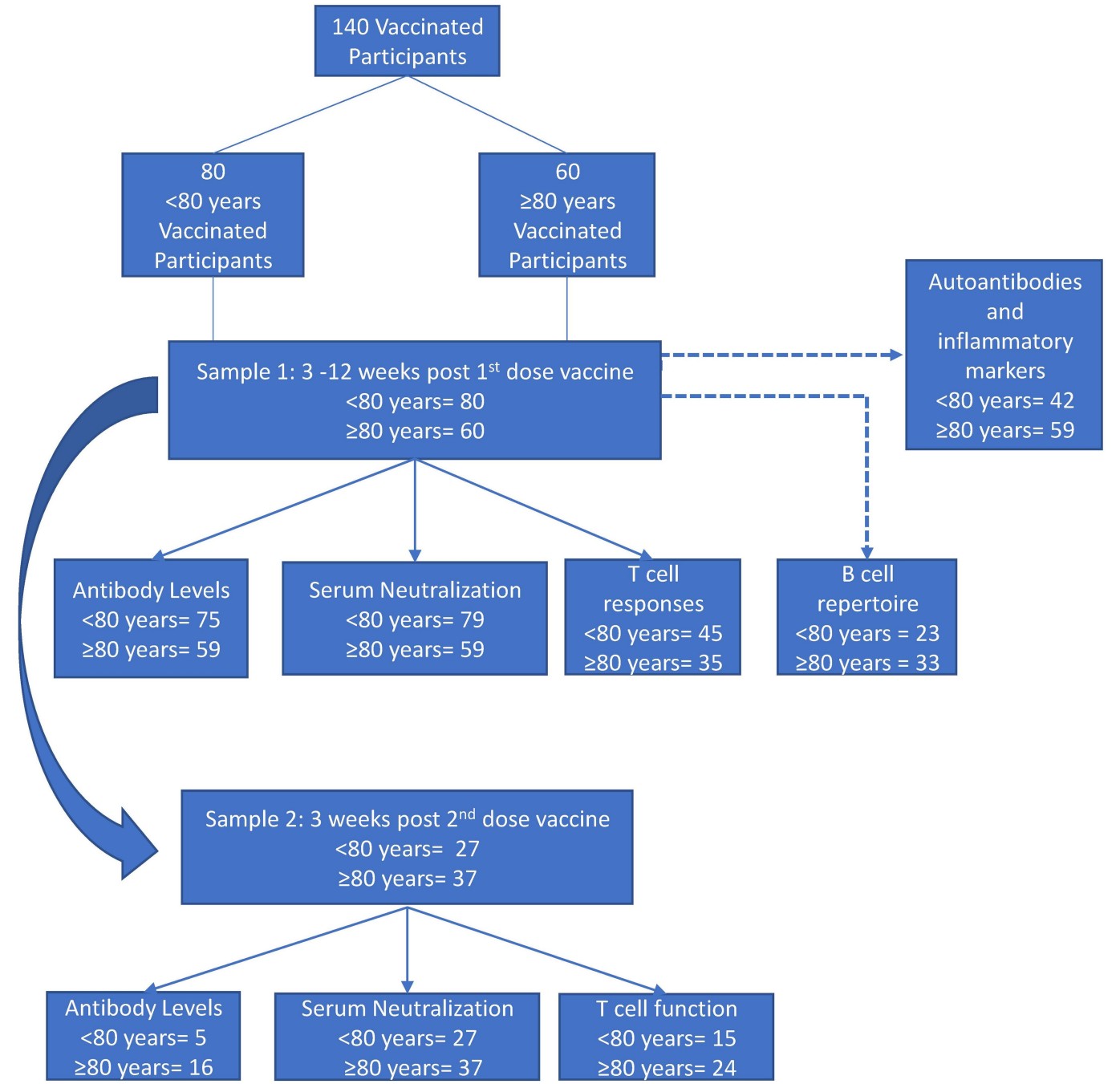

**Extended Data Fig. 1 | Study flow diagram for samples and analyses.** *n* values are shown for each analysis.

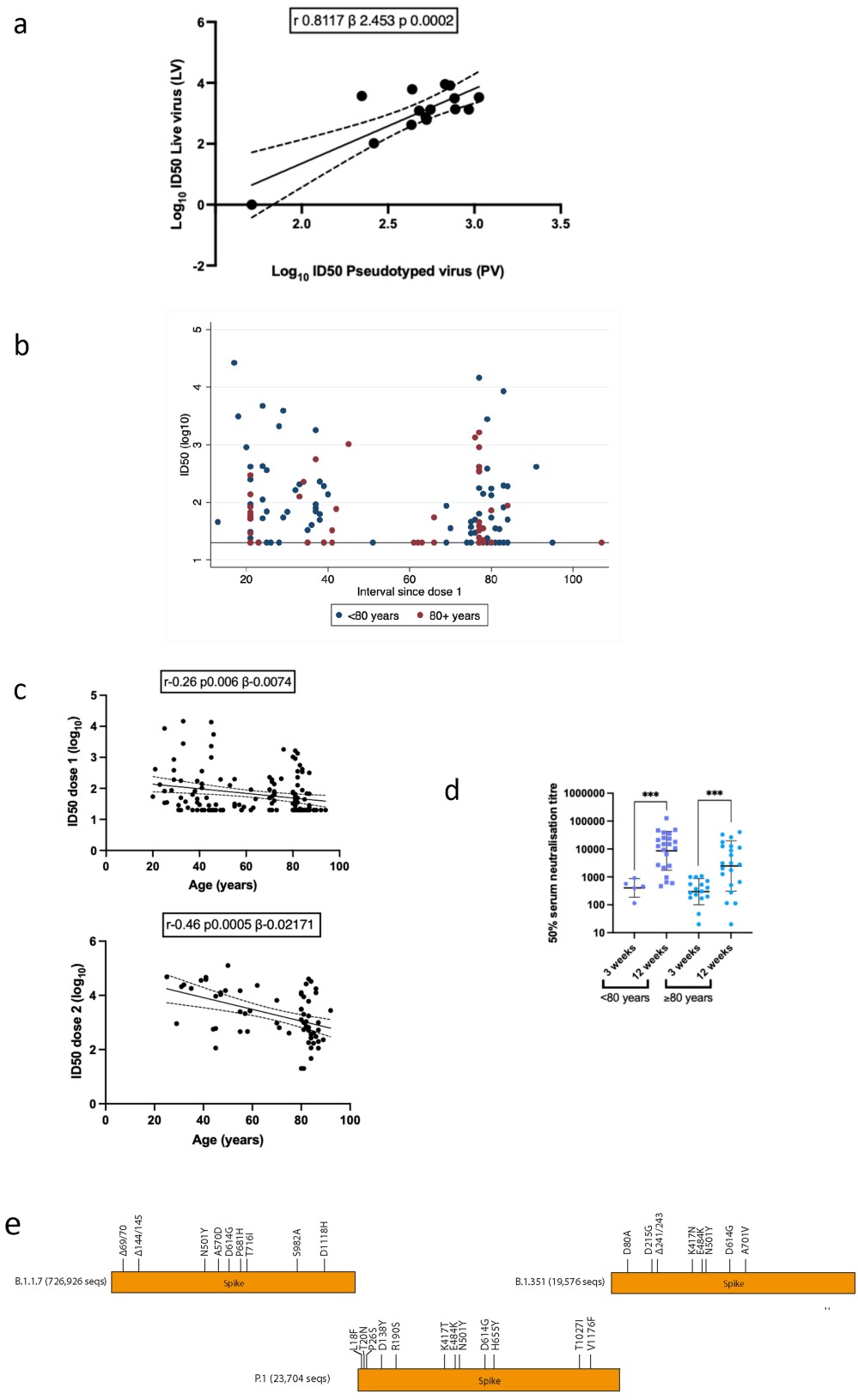

**Extended Data Fig. 2 | SARS-CoV-2 neutralization by serum from individuals vaccinated with Pfizer BNT162b2 vaccine. a**, Linear correlation of live virus neutralization with SARS-CoV-2 spike PV neutralization for 13 sera from individuals vaccinated with BNT162b2. Linear regression line plotted bounded by 95% CI. **b**, SARS-CoV-2 PV neutralization by sera from individuals vaccinated with first dose of BNT162b2 ($n = 140$) plotted against time since first dose. **c**, Correlation of SARS-CoV-2 neutralization by sera from individuals vaccinated with BNT162b2 with age. Serum neutralization of spike (D614G)

pseudotyped lentiviral particles (ID50) after dose 1 (top, $n = 138$) or dose 2 (bottom, $n = 64$) by age. Linear regression line plotted bounded by 95% CI. Bonferroni adjustment was made for multiple comparisons in linear regression. **d**, ID50 against wild-type (D614G) PV following the second dose of vaccine stratified by age and interval between vaccine doses (3 weeks ($n = 21$) and 12 weeks ($n = 43$)). GMT ± s.d., Mann–Whitney test. **e**, Spike mutations in VOCs, along with number of sequences in GISAID database.

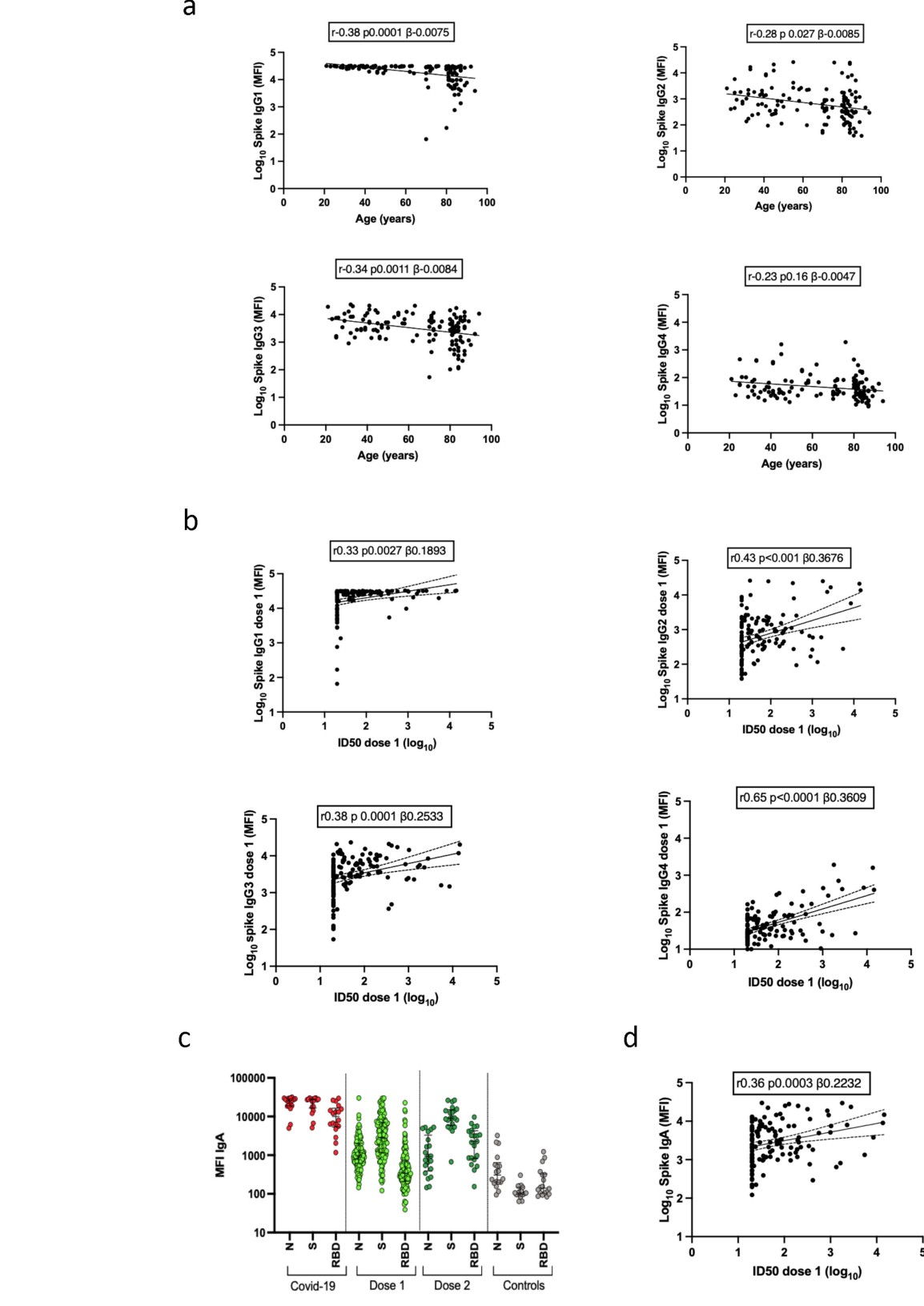

**Extended Data Fig. 3 | Binding IgG and IgA spike antibody responses following BNT162b2 vaccination. a**, Correlations between serum binding IgG subclass 1–4 antibody responses following vaccination with first dose of BNT162b2 and age in years (*n* = 133). **b**, Correlations between serum binding IgG subclass 1–4 antibody responses following vaccination with first dose of BNT162b2 and serum neutralization using a PV system (*n* = 133). **c**, IgA responses to spike, nucleocapsid and RBD after first dose (light green, *n* = 133) and second dose (dark green, *n* = 21) compared to individuals with prior infection (red, *n* = 18) and negative controls (grey, *n* = 18) at serum dilution of 1 in 100. **d**, Correlations between serum binding IgA spike antibody responses following vaccination with first dose of BNT162b2 and serum neutralization using a PV system (*n* = 133). Bonferroni adjustment was made for multiple comparisons. Spike proteins tested are Wuhan-1 with D614G (WT). Linear regression lines plotted bounded by 95% CI.

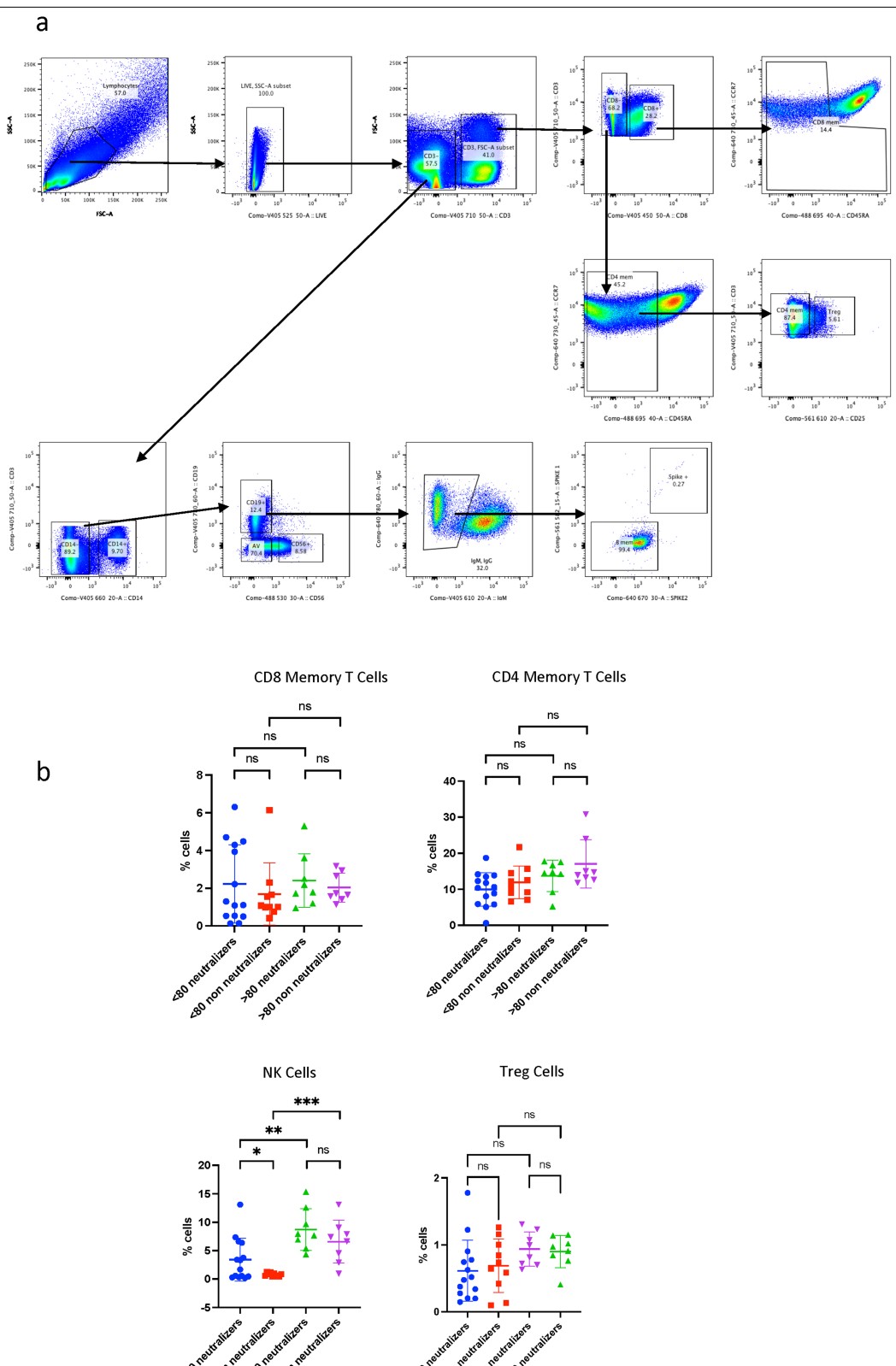

**Extended Data Fig. 4 | Peripheral blood lymphocyte subsets following first dose of BNT162b2.** PBMCs were sorted by FACS (*n* = 16 above 80 years of age, *n* = 16 below 80 years of age). **a**, Gating strategy for flow cytometry analysis of human immune cells after vaccination. **b**, Data for indicated sorted cell subsets stratified by neutralizing response after first dose (*n* = 8 in each category). NK cells, natural killer cells; Treg cells, regulatory T cells. Error bars, s.d.

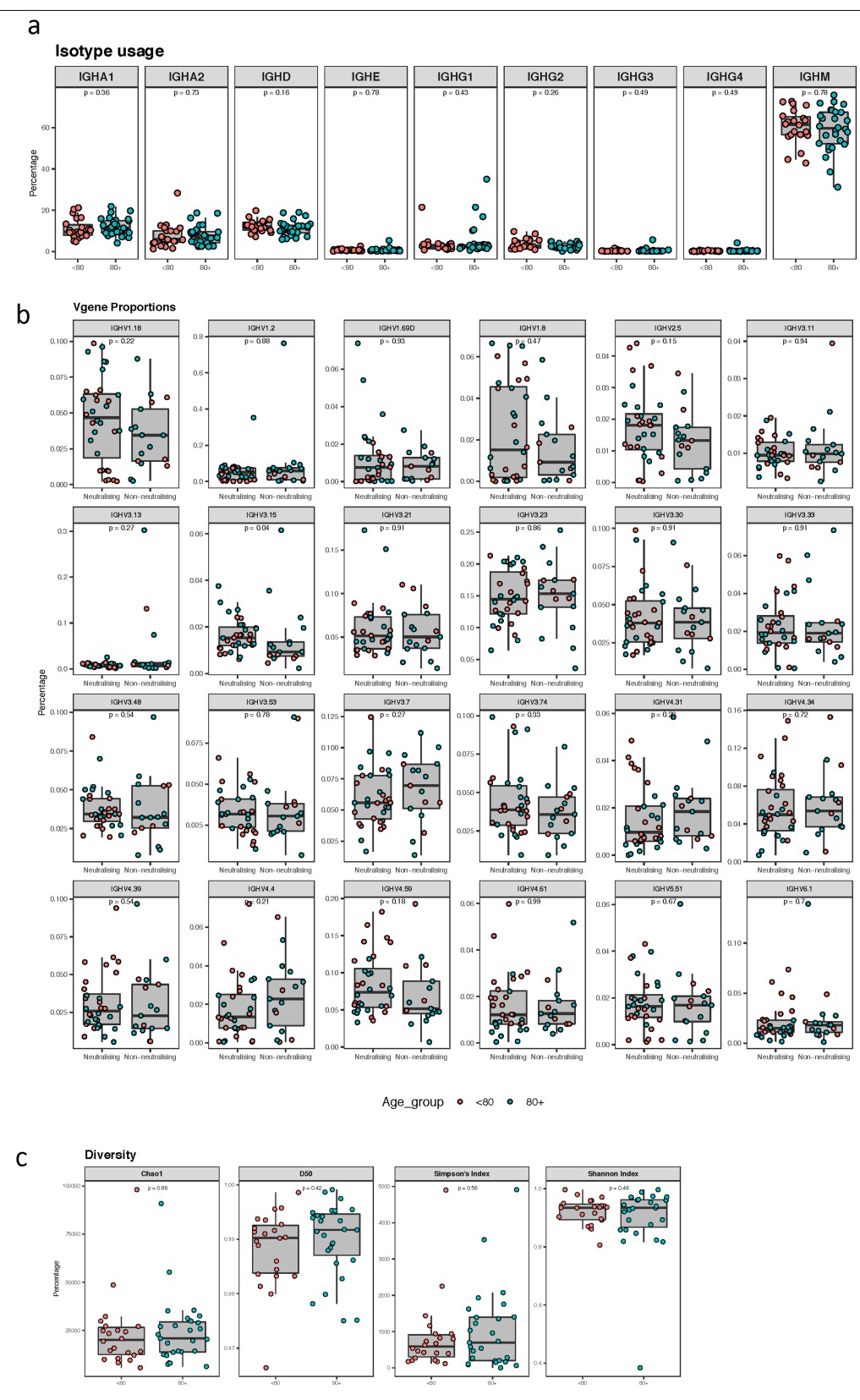

**Extended Data Fig. 5 | B cell repertoire following vaccination with first dose of BNT162b2. a**, Isotype usage according to unique VDJ sequence in <80-year-old ($n$ = 22) and ≥80-year-old groups ($n$ = 28). Differences between groups were calculated using Mann–Whitney $U$-test. **b**, V gene usage as a proportion, by neutralization of spike PV. Neutralization cut-off for 50% neutralization was set at 20. Differences between groups were calculated using Mann–Whitney $U$-test. **c**, Diversity indices comparing the two age groups. The inverse is depicted for Simpson's index and the Shannon–Weiner index is normalized. Differences between groups were calculated using a $t$-test. For boxplots: centre line, median; box, 25th–75th percentile; whiskers, 1.5× IQR.

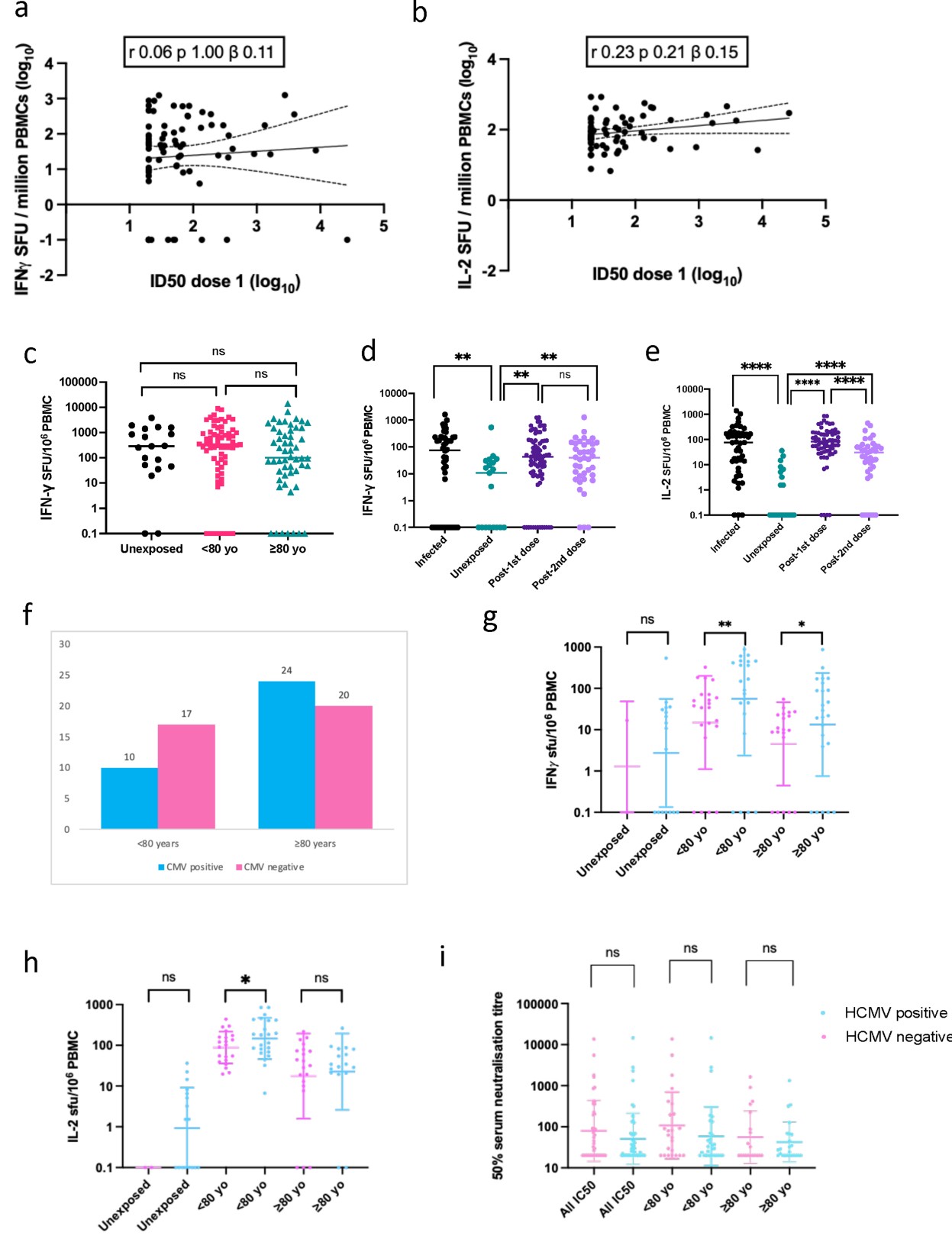

**Extended Data Fig. 6** | See next page for caption.

**Extended Data Fig. 6 | T cell responses following vaccination with BNT162b2.** Correlation between T cell responses against SARS-CoV-2 spike peptide pool and serum neutralization of spike (D614G) pseudotyped lentiviral particles (ID50). **a**, **b**, Correlation of IFNγ (**a**, $n = 79$) and IL-2 (**b**, $n = 69$) FluoroSpot and ID50 after first dose. Linear regression lines with 95% CI are plotted. Bonferroni adjustment was made for multiple comparisons. **c**, FluoroSpot IFNγ PBMC responses to peptide pool of CEF peptide pool. Responses from unexposed PBMCs (stored from 2014–2016, $n = 20$), <80 years group ($n = 46$) and ≥80 years group ($n = 35$) three weeks after the first dose of vaccine. **d**, **e**, FluoroSpot analysis for IFNγ (**d**) and IL-2 T cell responses (**e**) specific to SARS-CoV-2 spike protein peptide pool following stimulation of PBMCs from infected donors ($n = 46$), unexposed donors ($n = 20$) and vaccinated individuals three weeks or more after the first dose (IFNγ, $n = 77$; IL-2, $n = 64$) and three weeks after the second dose (IFNγ and IL-2, $n = 39$). **f**–**i**, Human cytomegalovirus serostatus, T cell responses and serum neutralization of spike (D614G) pseudotyped lentiviral particles (ID50) after the first dose of vaccine. **f**, HCMV serostatus for <80- and ≥80-year age groups ($n = 72$). **g**, **h**, IFNγ (**g**, $n = 72$) and IL-2 (**h**, $n = 64$) FluoroSpot responses after the first dose. **i**, ID50 after the first dose by CMV serostatus. Error bars, s.d.

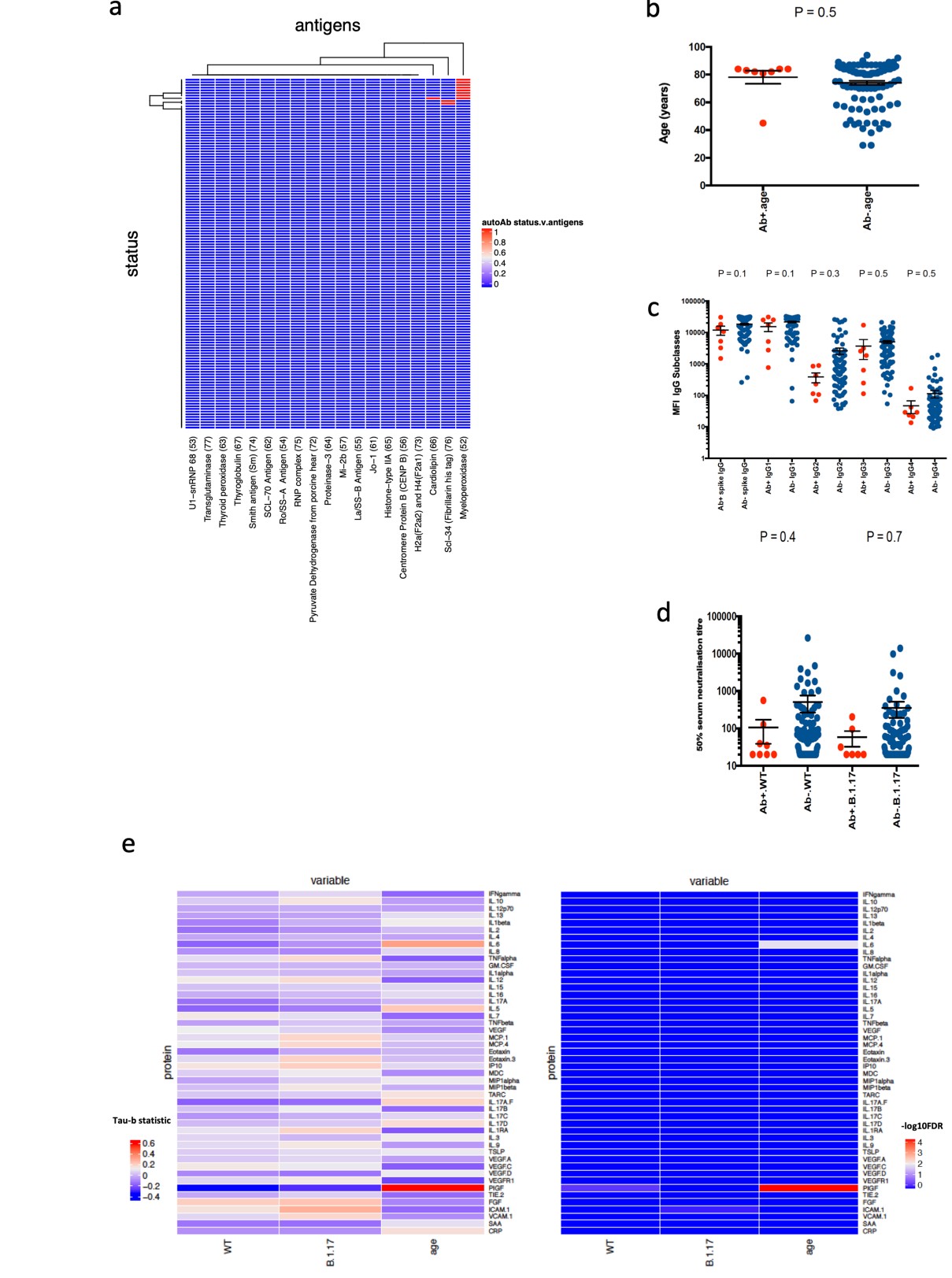

**Extended Data Fig. 7 | Autoantibodies and inflammatory markers in participants who received at least one dose of the BNT162b2 vaccine and relationship to SARS-CoV-2 spike-specific IgG and SARS-CoV-2 PV neutralization.** $n = 101$. **a**, Heatmap of $\log_2$-transformed fluorescence intensity of 19 autoantibodies; red, positive; blue, negative. **b**, Age (mean ± s.d.) in years by anti-MPO antibody-positive (red) or -negative (blue) status. **c**, IgG subclass responses to spike after first dose of BNT162b2 vaccine in individuals with or without anti-MPO antibodies ($n = 100$). **d**, GMT ± s.d of sera from individuals after their first dose of vaccine against wild-type and B.1.1.7 spike mutant SARS-CoV-2 PVs by anti-MPO antibody status. **e**, Nonparametric rank correlation (Kendall's tau-*b*) of wild-type (WT) PV neutralization, variant (B.1.17) PV neutralization and age (<80 or ≥80 years) against each of 53 cytokines or chemokines. Heatmaps illustrate Tau-*b* statistic (left) and significance (right, −$\log_{10}$FDR).

**Extended Data Table 1 | Characteristics of study participants and neutralization data for wild-type SARS-CoV-2**

| | <80 years (N=80 or n/N) | ≥80 years (N=60 or n/N) | P value |
|---|---|---|---|
| **Female %** | 60.0 (48) | 38.3 (23) | 0.01[a] |
| **Median age (IQR)** *years* | 45.5 (36.0-67.0) | 83.0 (81.0-85.5) | - |
| **Sera GMT WT (95% CI)** | | | |
| dose 1 | 104.1 (69.7-155.2)[b] | 48.2 (34.6-67.1)[c] | 0.004[d] |
| dose 2 | 4894.4 (2305.0-10392.6)[e] | 987.2 (506.4-1924.4)[f] | 0.003[d] |
| **Serum ID50<20 for WT %** | | | |
| dose 1 | 22.8 (18/79) | 50.9 (30/59) | 0.001[a] |
| dose 2 | 0 (0/27)[e] | 5.4 (2/37)[f] | 0.02[g] |
| **Prior SARS-CoV-2** | 6.8 (5/74) | 8.6 (5/58) | 0.69 |

[a]Chi-square test.
[b]Neutralization data unavailable for two individuals.
[c]Neutralization data unavailable for one individual.
[d]Mann–Whitney test.
[e]Neutralization data available for 27 of 80 individuals.
[f]Neutralization data available for 37 of 60 individuals.
[g]Test of proportions.

**Extended Data Table 2 | Neutralization after the first dose of BNT162b2 vaccine against wild-type and B1.1.7 PVs**

| | Number | Risk ID50<20 | Unadjusted OR (95% CI) | P value | Adjusted OR* (95% CI) | P value |
|---|---|---|---|---|---|---|
| *WT* | | | | | | |
| **Age group** *years* | | | | | | |
| <80 | 78 | 23.1 (18/78) | 1 | | 1 | |
| ≥ 80 | 59 | 50.9 (30/59) | 3.4 (1.7-7.2) | 0.001 | 3.7 (1.7-8.1) | 0.001 |
| **Sex** | | | | | | |
| Male | 68 | 32.4 (22/68) | 1 | | 1 | |
| Female | 69 | 37.7 (26/69) | 1.2 (0.6- 2.6) | 0.52 | 1.4 (0.6-3.3) | 0.39 |
| **Time since dose 1** *weeks* | | | | | | |
| 3-8 | 68 | 29.4 (20/68) | 1 | | 1 | |
| 9-12 | 69 | 40.6 (28/69) | 1.6 (0.8-3.3) | 0.17 | 1.6 (0.7-3.6) | 0.25 |
| **Previous COVID-19** | | | | | | |
| No | 121 | 35.5 (43/121) | 1 | | 1 | |
| Yes | 10 | 40.0 (4/100) | 1.2 (0.3-4.5) | 0.28 | 1.1 (0.3-4.6) | 0.92 |
| **B.1.1.7** | | | | | | |
| **Age group** *years* | | | | | | |
| <80 | 77 | 25.9 (20/77) | 1 | | 1 | |
| ≥ 80 | 58 | 60.3 (35/58) | 4.3 (2.1-9.0) | <0.001 | 4.3 (2.0-9.3) | <0.001 |
| **Sex** | | | | | | |
| Male | 67 | 43.3 (29/67) | 1 | | 1 | |
| Female | 68 | 38.2 (26/68) | 0.8 (0.4-1.6) | 0.55 | 1.2 (0.6-2.8) | 0.59 |
| **Time since dose 1** *weeks* | | | | | | |
| 3-8 | 66 | 42.4 (28/66) | 1 | | 1 | |
| 9-12 | 69 | 39.1 (27/69) | 0.9 (0.4-1.7) | 0.70 | 0.7 (0.3-1.6) | 0.41 |
| **Previous COVID-19** | | | | | | |
| No | 120 | 41.7 (50/120) | 1 | | 1 | |
| Yes | 10 | 40.0 (4/10) | 0.9 (0.3-3.5) | 0.92 | 0.9 (0.2-3.6) | 0.88 |

*Mutually adjusted for other variables in the table. OR, odds ratio.

**Extended Data Table 3 | Neutralization in participants after the first dose of BNT162b2 vaccine against wild-type and B.1.1.7, B.1.351 and P.1 spike mutant PVs**

| | Number | Risk ID50<20 | Unadjusted OR (95% CI) | P value | Adjusted OR (95% CI) | P value |
|---|---|---|---|---|---|---|
| **WT** | | | | | | |
| **Age group** *years* | | | | | | |
| <80 | 55 | 25.5 (14/55) | 1 | | 1 | |
| ≥ 80 | 27 | 48.2 (13/27) | 2.7 (1.3-7.2) | 0.04 | 2.4 (0.8-6.8) | 0.06 |
| **Sex** | | | | | | |
| Male | 33 | 33.3 (11/33) | 1 | | 1 | |
| Female | 49 | 32.7 (16/49) | 1.0 (0.4-2.5) | 0.95 | 1.0 (0.3-3.2) | 0.94 |
| **Time since dose 1** *weeks* | | | | | | |
| 3-8 | 28 | 25.0 (7/28) | 1 | | 1 | |
| 9-12 | 54 | 37.0 (20/54) | 1.8 (0.6-4.9) | 0.27 | 1.9 (0.6-6.0) | 0.25 |
| **Previous COVID-19** | | | | | | |
| No | 72 | 33.3 (24/72) | 1 | | 1 | |
| Yes | 6 | 50.0 (3/6) | 1.7 (0.4-10.7) | 0.42 | 1.8 (0.3-10.4) | 0.53 |
| **B.1.1351** | | | | | | |
| **Age group** *years* | | | | | | |
| <80 | 55 | 69.1 (38/55) | 1 | | 1 | |
| ≥ 80 | 27 | 81.5 (22/27) | 2.0 (0.6- 6.1) | 0.24 | 1.7 (0.5-5.7) | 0.41 |
| **Sex** | | | | | | |
| Male | 33 | 72.7 (24/33) | 1 | | 1 | |
| Female | 49 | 73.5 (36/49) | 1.0 (0.4-2.8) | 0.94 | 1.3 (0.4- 4.4) | 0.66 |
| **Time since dose 1** *weeks* | | | | | | |
| 3-8 | 28 | 75.0 (21/28) | 1 | | 1 | |
| 9-12 | 54 | 72.2 (39/54) | 0.9 (0.3-2.5) | 0.79 | 1.0 (0.3-3.4) | 0.94 |
| **Previous COVID-19** | | | | | | |
| No | 72 | 77.8 (56/72) | 1 | | 1 | |
| Yes | 6 | 66.7 (4/6) | 0.6 (0.1- 3.4) | 0.54 | 0.5 (0.1-3.3) | 0.51 |
| **P.1** | | | | | | |
| **Age group** *years* | | | | | | |
| <80 | 55 | 52.7 (29/55) | 1 | | 1 | |
| ≥ 80 | 27 | 88.9 (24/27) | 7.2 (1.9-26.6) | 0.003 | 6.7 (1.7- 26.3) | 0.008 |
| **Sex** | | | | | | |
| Male | 33 | 66.7 (22/33) | 1 | | 1 | |
| Female | 49 | 63.3 (31/49) | 0.9 (0.3-2.2) | 0.75 | 1.2 (0.4-4.0) | 0.71 |
| **Time since dose 1** *weeks* | | | | | | |
| 3-8 | 28 | 60.7 (17/28) | 1 | | 1 | |
| 9-12 | 54 | 66.7 (36/54) | 1.3 (0.5-3.3) | 0.59 | 1.5 (0.5-4.7) | 0.46 |
| **Previous COVID-19** | | | | | | |
| No | 72 | 68.1 (49/72) | 1 | | 1 | |
| Yes | 6 | 66.7 (4/6) | 0.9 (0.2-5.5) | 0.94 | 0.8 (0.1-5.5) | 0.77 |

*Mutually adjusted for other variables in the table.

# nature research

# Reporting Summary

Nature Research wishes to improve the reproducibility of the work that we publish. This form provides structure for consistency and transparency in reporting. For further information on Nature Research policies, see Authors & Referees and the Editorial Policy Checklist.

## Statistics

For all statistical analyses, confirm that the following items are present in the figure legend, table legend, main text, or Methods section.

| n/a | Confirmed | |
|---|---|---|
| ☐ | ☒ | The exact sample size (*n*) for each experimental group/condition, given as a discrete number and unit of measurement |
| ☐ | ☒ | A statement on whether measurements were taken from distinct samples or whether the same sample was measured repeatedly |
| ☐ | ☒ | The statistical test(s) used AND whether they are one- or two-sided<br>*Only common tests should be described solely by name; describe more complex techniques in the Methods section.* |
| ☐ | ☒ | A description of all covariates tested |
| ☐ | ☒ | A description of any assumptions or corrections, such as tests of normality and adjustment for multiple comparisons |
| ☐ | ☒ | A full description of the statistical parameters including central tendency (e.g. means) or other basic estimates (e.g. regression coefficient) AND variation (e.g. standard deviation) or associated estimates of uncertainty (e.g. confidence intervals) |
| ☐ | ☒ | For null hypothesis testing, the test statistic (e.g. *F*, *t*, *r*) with confidence intervals, effect sizes, degrees of freedom and *P* value noted<br>*Give P values as exact values whenever suitable.* |
| ☒ | ☐ | For Bayesian analysis, information on the choice of priors and Markov chain Monte Carlo settings |
| ☒ | ☐ | For hierarchical and complex designs, identification of the appropriate level for tests and full reporting of outcomes |
| ☐ | ☒ | Estimates of effect sizes (e.g. Cohen's *d*, Pearson's *r*), indicating how they were calculated |

*Our web collection on statistics for biologists contains articles on many of the points above.*

## Software and code

Policy information about availability of computer code

| Data collection | Graphpad Prism v9 was used to produce figures.<br>Stata v13 was used for statistical analyses.<br>FlowJo version 10.7.1 for flow cytometry analyses.<br>IMGT-V QUEST was used for immunoglobulin gene use and sequence annotation<br>B cell receptor repertoire analyses was performed in Python. |
|---|---|
| Data analysis | Logistic regression was used to model the association between age group and neutralisation by vaccine-elicited antibodies after the first dose of the BNT162b2 vaccine. The effect of sex and time interval from vaccination to sampling as confounders were adjusted for. Linear regression was also used to explore the association between age as a continuous variable and log transformed ID50, binding antibody levels, antibody subclass levels and T cell response after dose 1 and dose 2 of the BNT162b2 vaccine.<br><br>The difference in continuous and categorical data were tested using Wilcoxon rank sum or Mann-Whitney test and Chi square test respectively. |

For manuscripts utilizing custom algorithms or software that are central to the research but not yet described in published literature, software must be made available to editors/reviewers. We strongly encourage code deposition in a community repository (e.g. GitHub). See the Nature Research guidelines for submitting code & software for further information.

## Data

Policy information about availability of data

All manuscripts must include a data availability statement. This statement should provide the following information, where applicable:

- Accession codes, unique identifiers, or web links for publicly available datasets
- A list of figures that have associated raw data
- A description of any restrictions on data availability

Sequence data have been deposited at the European Genome -phenome Archive (EGA) which is hosted by the EBI and the CRG under accession number EGAS00001005380.

# Field-specific reporting

Please select the one below that is the best fit for your research. If you are not sure, read the appropriate sections before making your selection.

☒ Life sciences    ☐ Behavioural & social sciences    ☐ Ecological, evolutionary & environmental sciences

For a reference copy of the document with all sections, see nature.com/documents/nr-reporting-summary-flat.pdf

# Life sciences study design

All studies must disclose on these points even when the disclosure is negative.

| | |
|---|---|
| Sample size | We assumed a risk ratio of non-neutralisation in the ≥80 years group compared with <80 years group of 5. Using an alpha of 0.05 and power of 90% required a sample size of 50 with a 1:1 ratio in each group. We however recruited 140 participants in order to reduce the risk of Type I error as multiple analyses were undertaken. |
| Data exclusions | Consecutively presenting participants were recruited with no exclusion. |
| Replication | Experiments were done in technical duplicates and a repeat was done. |
| Randomization | Not applicable as this was not an intervention study. |
| Blinding | Not applicable as this was not an intervention study. |

# Reporting for specific materials, systems and methods

We require information from authors about some types of materials, experimental systems and methods used in many studies. Here, indicate whether each material, system or method listed is relevant to your study. If you are not sure if a list item applies to your research, read the appropriate section before selecting a response.

### Materials & experimental systems

| n/a | Involved in the study |
|---|---|
| ☐ | ☒ Antibodies |
| ☐ | ☒ Eukaryotic cell lines |
| ☒ | ☐ Palaeontology |
| ☒ | ☐ Animals and other organisms |
| ☐ | ☒ Human research participants |
| ☒ | ☐ Clinical data |

### Methods

| n/a | Involved in the study |
|---|---|
| ☒ | ☐ ChIP-seq |
| ☐ | ☒ Flow cytometry |
| ☒ | ☐ MRI-based neuroimaging |

## Antibodies

| | |
|---|---|
| Antibodies used | anti-CD4+ or anti-CD8+ direct beads (Miltenyi Biotec)<br>CD3-FITC, CD4-PE, and CD8- PerCPCy5.5 antibody (BioLegend)<br>CD19 (SJ25C, 363028), CD3 (OKT3, 317328), CD11c (3.9, 301608), CD25 (M-A251, 356126), CD14 (M5E2,301836), and IgM (IgG1-k, 314524) (all Biolegend).<br>CCR7 (150503, 561143) and IgG (G18-145, 561297) (BD Biosciences)<br>CD45RA (T6D11, 130-113-359) ( Miltyeni Biotech)<br>CD8a (SK1, 48-0087-42) (eBiosciences) |
| Validation | Validation by manufacturer as detailed on their website. |

# Eukaryotic cell lines

Policy information about cell lines

| | |
|---|---|
| Cell line source(s) | HEK 293T and HeLa cells were used. |
| Authentication | None of the cell lines used were authenticated. |
| Mycoplasma contamination | All cell lines used were tested (by PCR) and were mycoplasma free. |
| Commonly misidentified lines (See ICLAC register) | No commonly misidentified lines were used in this study. |

# Human research participants

Policy information about studies involving human research participants

| | |
|---|---|
| Population characteristics | Community participants or health care workers receiving the first dose of the BNT162b2 vaccine between the 14th of December 2020 to the 10th of February 2021 were consecutively recruited. 140 participants were recruited. |
| Recruitment | Participants attending Addenbrooke's Hospital for their COVID-19 vaccination were recruited through the NIHR BioResource Centre Cambridge. |
| Ethics oversight | The study was approved by the East of England – Cambridge Central Research Ethics Committee. |

Note that full information on the approval of the study protocol must also be provided in the manuscript.

# Flow Cytometry

## Plots

Confirm that:

☒ The axis labels state the marker and fluorochrome used (e.g. CD4-FITC).

☒ The axis scales are clearly visible. Include numbers along axes only for bottom left plot of group (a 'group' is an analysis of identical markers).

☒ All plots are contour plots with outliers or pseudocolor plots.

☒ A numerical value for number of cells or percentage (with statistics) is provided.

## Methodology

| | |
|---|---|
| Sample preparation | PBMCs were isolated from whole blood samples using Lymphoprep and stored in liquid nitrogen. Cells were thawed and stained in PBS containing 2nM EDTA at 4 °C |
| Instrument | Instrument: FACSAria Fusion (BD Biosciences). |
| Software | FlowJo version 10.7.1 |
| Cell population abundance | 2 x 10^5 to 2 x 10^6 |
| Gating strategy | Total lymphocytes were gated, then live dead. From live dead, CD3- and CD3+ cells were separated. CD3+ cells were separated into CD8- and CD8+ cells. CD8+ memory cells were determined using CCR7 and CD45RA from the CD8+ gate. From the CD8- gate, CD4 memory cells were determined using CCR7 and CD45RA. From the CD4 memory gate, Tregs were separated from CD4 memory cells with CD25. The CD3- gate was used to separate CD14- and CD14+ cells. CD14- cells were separated in CD19+CD56- and CD19-CD56+ gates. From the CD19+ gate, B memory cells were determined using IgG and IgM. IgG+IgM- cells were gated and from these cells, B memory cells doubly positive for Spike were gated. |

☒ Tick this box to confirm that a figure exemplifying the gating strategy is provided in the Supplementary Information.

