## [Peer Review File · Nature]

Manuscript Title: Age-related heterogeneity in immune responses to SARS-CoV-2 following BNT162b2 vaccination

Reviewer Comments & Author Rebuttals

Reviewer Reports on the Initial Version:

Referees' comments:

Referee #1 (Remarks to the Author):

The authors present some of the only data available to date on SARS-CoV-2 mRNA vaccine immunogenicity in an elderly cohort (n=101), with a median age of 81 (IQR 70-84). The majority of data presented are following one dose of the BNT162b2 vaccine, with a smaller number (n=21) having received two doses. The primary findings are those comparing pseudovirus neutralisation titres (to both D614G and B.1.1.7 spike constructs) in those <80 years compared to those 80 and above; with a geometric mean neutralisation titre to D614G of 83.4 (95% CI 52.0 - 133.7) in those under 80 vs 46.6 (95% CI 33.5-64.8) in the 80+ group after one dose. The OR (adjusted for sex and time from the 1st dose) between groups is reported as 3.7 (1.2-11.2), p=0.02. Neutralisation titres in both age groups were found to be lower to B.1.1.7 than D614G spike pseudotyped viruses. The authors also used ex vivo IFN-gamma/IL-2 Fluorospot assays to interrogate differences in spike-specific T-cell response across age groups and first/second doses. The main findings are that the second dose results in a boost to IFN-gamma producing T-cells elicited after the first dose in both age groups (<80 and 80+), but no significant increase in IL-2 producing T-cells. Following the first vaccine dose, no difference in IFN-gamma T-cell response was seen in individuals <80 compared to those 80 and over, but a lower IL-2 T-cell response was seen in the 80+ age group. There are a number of other data presented of varying relevance, novelty and interest, including correlation of binding antibody responses with neutralisation titres, levels of somatic hypermutation and levels of autoantibodies (eg anti-MPO and anti-cardiolipin).

Overall, the authors should be commended for gathering data in an elderly cohort, as there are very few manuscripts with such data for SARS-CoV-2 vaccine responses (e.g. Anderson et al, NEJM, 2020 for mRNA-1273; Li et al Nature Medicine, 2021 for BNT162b1; Parry et al, 2021, SSRN preprint comparing BNT162b2 and ChAdOx1, as well as the subset in the Phase I Walsh et al study in NEJM). The Li et al study in Chinese adults should be referenced and was likely published after submission of the current manuscript. While it is using the BNT162b1 vaccine, it includes data from 72 adults aged 65-85, along with an appropriate control group of adults aged 18-55 in the context of a double blind phase I immunogenicity study (live virus neutralisation, binding antibody and IFN-gamma T-cell responses).

However, other than the pseudovirus neutralisation data which are robust and useful, other aspects of the manuscript either have some methodological or statistical concerns and/or feel like they have been added in to expand the data to a full manuscript without any particular utility or focused and thorough immunological exploration of what may underlie the poorer antibody responses observed. Specific comments are given below:

1. The group presented that are <80 years old (n=42 of 101) have quite a wide age range and are still relatively old. Immunosenescence probably starts earlier than 80 years of age. To be a true representation of the title ('Age-related heterogeneity..'), the authors should ideally have included adequate numbers of participants <65 years, 65 - 79 and 80+ years.

2. Are there any details on previous SARS-CoV-2 exposure in the cohort? I could not see that this was clearly presented and as several publications now show, prior infection results in a significantly higher antibody and T-cell response following the first dose of vaccine. Reassurance is needed that there is no imbalance in the % of individuals with prior exposure in groups (eg higher in younger age

group), even if this is done based on anti-N antibodies in post-vaccine samples if pre-vaccine samples are not available. If this approach is used, then an extensively validated antibody assay used in clinical diagnostic laboratories should be utilised. There is of course the caveat about waning anti-N antibodies, so pre-vaccine samples are idea, but I appreciate this may not be available.

3. There is considerable heterogeneity in the sampling period after the 1st dose (3 - 8 weeks), which Fig 1C demonstrates. The authors have rightly used logistic regression to try and adjust for this. The Adjusted ORs and p values should be highlighted when reporting the differences between age groups throughout the abstract/manuscript.

4. It was not clear whether there was variability in the dosing interval in the 21 individuals whose post-2nd dose data are included. The authors state that the 3 week interval changed to 12 weeks in the middle of the study. This is important as there may be significant difference in the post-boost antibody titre as has been seen for the ChAdox vaccine.

5. There are very few individuals with post 2nd dose samples (21) and stratifying further into age <80 for 80+ (5 vs 16) makes many statistical comparisons the authors have carried out meaningless. For example it is difficult to interpret the non-significant result in neutralising titres between the two age groups.

6. The authors state that they were interested in seeing if the 'poorer' responses persisted up to 12 weeks after the 1st dose. It is not clear whether the samples used to answer this question were longitudinal samples (i.e. repeat samples) taken from individuals who had samples taken earlier after vaccination. If it is not and these are different individuals, it would not be appropriate to use these data to compare 'early vs late responses' given the small sample size and inter-individual heterogeneity in antibody responses seen.

7. The authors state the "Age showed statistically significant correlation with serum neutralisation of WT virus after the first but not second dose". Given the small sample size following the 2nd dose, this is not appropriate to conclude. The R is pretty much the same (-0.25 vs -0.29) but it is just that the p value is 0.28 potentially due to lack of power.

8. Fig 1F - please make it clear what the red and blue neutralisation curves represent.

9. The authors state that "IgA concentrations was significantly correlated with age (Figure 2B)." It should be made clear if this is after 1 dose, which would be the most appropriate comparison. Though again the paucity of individuals at younger age groups makes this comparison difficult to interpret in the context of a correlation.

10. The authors state that "IgG1-4 levels after dose 1 correlated strongly with neutralisation". I do not think that an R of 0.27 and 0.4 (Fig 2C and D) can be called a 'strong' correlation.

11. There are 24 comparisons made in Figure 3B looking at skewing of V gene use. While in the statistical methods the authors state that Bonferroni correction was performed to adjust for multiple comparisons, it is not clear in this figure or legend that this was done. Please confirm if the p values presented are following Bonferroni adjustment as if not it is possible that at least some of the significant results would be due to chance.

12. The T-cell results section is labelled as "CD4 and CD8 T cell responses to spike following mRNA vaccination". This is misleading as nowhere are responses stratified by CD4 and CD8. Given the differences in IL-2 and IFN-g across doses and age groups the authors report, the manuscript would benefit significantly if there were ICS data (or CD4/CD8 depleted ELISpot data) showing whether these differences were due to differences in spike-specific CD4 and CD8 induction across ages or doses. This is a major deficiency in my opinion.

13. There appears to be quite high background in the IFNg assays – surprising that even in <80s, the assay used does not pick up many T-cell responses after a single dose, which is in contrast to some other studies. This may be due to some methodological issues.

a) In some cases significantly less PBMCs appear to have been used (100,000 cells/well). The

responses seen with this may be significantly different and unable to detect low level responses compared to using 250,000 cells/well (a more usual number). This may be a significant methodological weakness – particularly if there was any bias in how many in each group were used with lower vs higher cell numbers.

b) There are no details on overlapping peptides for spike – whole spike? What length of peptides, what was the overlap?

c) Why did the authors choose 1mcg/ml as a final concentration to stimulate with in T-cell assays? Many assays/publications have used 2mcg/mL (or in some cases 10mcg/mL – eg the ChadOx vaccine immunogenicity data from clinical trials). This could have affected the magnitude of responses seen.

14. The age stratified statistical comparisons with the <80 post 2nd dose T-cell data seem a little meaningless with 5 data points.

15. I am not sure that the autoantibody data add much to this manuscript.

Referee #2 (Remarks to the Author):

Collier and colleagues report on a cohort of 101 individuals of median age 81 years that received 1 or 2 doses of SARS-CoV-2 mRNA vaccine (BTN162b2). The group was split into those that are 80 or older, and those that were younger than 80 for analysis. Samples from participants were analyzed for neutralizing activity against Wuhan and the B.1.17 variant, ELISA binding titer, V gene usage, somatic mutation and T cell cytokine responses. They find that the people that were over 80 had lower neut responses after 1 dose of vaccine but caught up after the second dose. That as many as 55-57.8% of the older individuals receiving a single vaccine had measurable neut titers against B.1.1.7. They performed bulk BCR sequencing and found some changes in VH representation between the 2 groups but these were not correlated to neut activity which would have required antibody cloning and expression. Small differences were seen in somatic mutation. T cell responses to spike were measured by INFg and IL2 spot assays.

This an important topic, and one that has already been reported on by others. The key result that responses are generally lower after one than after 2 immunizations is well established. Moreover, it has also been reported that (Refs3 and 19) older individuals are most likely to show only low or no detectable levels of neut activity after the first dose. Whether this correlates with protection from hospitalization, which is the key parameter, is not clear and is clarified by this report.

Overall, the analysis is superficial for both B and T cell responses. On the B cell side antibody cloning and expression would be required to further understand what happened and on the T cell side much broader analysis of phenotype and cytokine responses including single cell mRNA analysis would be important. In addition, it is not at all clear why they separate individuals into 2 groups at age 80. Why not analyze on a continuum to see if there is an age cut off for the effects they report.

Referee #3 (Remarks to the Author):

The authors have put forth significant effort and the apparent intent of this report could be of great value.

The paper would benefit from some significant modifications. The (2-3) key points are difficult to determine from the abstract. The "study" design is not apparent in the abstract and even in the text it is confusing to follow.

The key points (whatever the authors determine to be key) should be clear in abstract, results, and then discussed, explained, and their significance explained in the discussion.

The discussion currently seems to ramble and include multiple only somewhat relevant facts (and some that seem less relevant), difficult to interpret implied implications for these findings, and the message is not clear or cohesive, so it is difficult to read and also to take away key messages.

Specific comments:

Please include data related to age as a continuous variable, an arbitrary cutoff of 80 years has minimal value to public health operations---if in fact a continuous variable analysis were used it may show that the drop off in response after 1st dose occurs closer to 60 or 70 years, or is more extreme at 85-90 years. Here the reader is left to wonder but certainly there is nothing magical about the immune response at the age of 80 as compared to 75 or 85.

Seems the key point of the paper is that older individuals do not mount a clinically significant response (at least if Neut Ab turns out to be the Correlate of Protection (CoP) for mRNA) after a single dose of mRNA. This is key and should be fine tuned in the message and results.

It is also important for the authors to clarify to the reader that the CoP is unknown. It may not be neutralizing antibody and rather may be binding antibody---if so, the premise of this report is flawed. That unknown should be declared up front and again in the discussion very clearly.

As for T cell responses, describing the quality of response without context to the reader is not helpful. There is no known T cell mechanism of protection for these vaccines and thus the intent of the data here are even more speculative. The authors could elaborate on the meaning of the different types of T cell responses but clarify the value is is speculative for this vaccine/infection.

Multiple speculative statements need to be removed or explained as such---related to speculation about the presumed response against the 351 variant or "escape variants". There is no data here and this is speculative.

The ref to high dose influenza is odd---is the author suggesting high dose SARS CoV2 vaccine is needed?

The ref to HIV vaccine responses is not useful at least as written---HIV vaccines have not been effective and so the immune responses to ineffective investigational vaccines are not a benchmark for comparing to a known effective vaccine---this info and ref seems out of place and not useful.

The speculation in the last of the discussion that those who respond poorly to 1st injection (but do respond to 2nd) will have diminished durability is not founded in data here or elsewhere and should be removed as a speculation. Could be stated as a question to be answered.

Overall, there is a lot of important data here, but the story and key messages are not quite there as written for a top tier journal publication intended for broad audience.

Author Rebuttals to Initial Comments:

Reviewer comments and responses

The authors present some of the only data available to date on SARS-CoV-2 mRNA vaccine immunogenicity in an elderly cohort (n=101), with a median age of 81 (IQR 70-84). The majority of data presented are following one dose of the BNT162b2 vaccine, with a smaller number (n=21) having received two doses. The primary findings are those comparing pseudovirus neutralisation titres (to both D614G and B.1.1.7 spike constructs) in those <80 years compared to those 80 and above; with a geometric mean neutralisation titre to D614G of 83.4 (95% CI 52.0 - 133.7) in those under 80 vs 46.6 (95% CI 33.5-64.8) in the 80+ group after one dose. The OR (adjusted for sex and time from the 1st dose) between groups is reported as 3.7 (1.2-11.2), p=0.02. Neutralisation titres in both age groups were found to be lower to B.1.1.7 than D614G spike pseudotyped viruses. The authors also used ex vivo IFN-gamma/IL-2 Fluorospot assays to interrogate differences in spike-specific T-cell response across age groups and first/second doses. The main findings are that the second dose results in a boost to IFN-gamma producing T-cells elicited after the first dose in both age groups (<80 and 80+), but no significant increase in IL-2 producing T-cells. Following the first vaccine dose, no difference in IFN-gamma T-cell response was seen in individuals <80 compared to those 80 and over, but a lower IL-2 T-cell response was seen in the 80+ age group. There are a number of other data presented of varying relevance, novelty and interest, including correlation of binding antibody responses with neutralisation titres, levels of somatic hypermutation and levels of autoantibodies (eg anti-MPO and anti-cardiolipin).

Overall, the authors should be commended for gathering data in an elderly cohort, as there are very few manuscripts with such data for SARS-CoV-2 vaccine responses (e.g. Anderson et al, NEJM, 2020 for mRNA-1273; Li et al Nature Medicine, 2021 for BNT162b1; Parry et al, 2021, SSRN preprint comparing BNT162b2 and ChAdOx1, as well as the subset in the Phase I Walsh et al study in NEJM). The Li et al study in Chinese adults

should be referenced and was likely published after submission of the current manuscript. While it is using the BNT162b1 vaccine, it includes data from 72 adults aged 65-85, along with an appropriate control group of adults aged 18-55 in the context of a double blind phase I immunogenicity study (live virus neutralisation, binding antibody and IFN-gamma T-cell responses).

Response: we thank the reviewer for appreciating the effort we have made to bring these data together, as suboptimal immune responses are likely to be a long term problem for control of SARS-CoV-2 and may exacerbate emergence of novel hyper mutated variants. These data hopefully should drive the search for ways of boosting immunity following vaccination. The recent data on B.1.617 transmissions in vaccinated individuals shows us that the elderly continue to be vulnerable in the post vaccine era.

However, other than the pseudovirus neutralisation data which are robust and useful, other aspects of the manuscript either have some methodological or statistical concerns and/or feel like they have been added in to expand the data to a full manuscript without any particular utility or focused and thorough immunological exploration of what may underlie the poorer antibody responses observed. Specific comments are given below:

Response: we thank the reviewer for this comment and would like to add that we have also generated data on B.1.351 and P.1 viruses across age groups in order to make the work translatable to the evolving situation globally.

1. The group presented that are <80 years old (n=42 of 101) have quite a wide age range and are still relatively old. Immunosenescence probably starts earlier than 80 years of age. To be a true representation of the title ('Age-related heterogeneity..'), the authors should ideally have included adequate numbers of participants <65 years, 65 - 79 and 80+ years.

Response: we are presently generating data on a further 37 individuals in the <65 age group.

2. Are there any details on previous SARS-CoV-2 exposure in the cohort? I could not see that this was clearly presented and as several publications now show, prior infection results in a significantly higher antibody and T-cell response following the first dose of vaccine. Reassurance is needed that there is no imbalance in the % of individuals with prior exposure in groups (eg higher in younger age group), even if this is done based on anti-N antibodies in post-vaccine samples if pre-vaccine samples are not available. If this approach is used, then an extensively validated antibody assay used in clinical diagnostic laboratories should be utilised. There is of course the caveat about waning anti-N antibodies, so pre-vaccine samples are idea, but I appreciate this may not be available.

Response: we have data on N antibodies that signifies prior exposure and have identified individuals with probable previous SARS-CoV-2 infection; 4 in the <80 years group and 5 in the ≥ 80 years group. We have performed sensitivity analyses by including and then removing these individuals from the analyses. The results did not change.

3. There is considerable heterogeneity in the sampling period after the 1st dose (3 - 8 weeks), which Fig 1C demonstrates. The authors have rightly used logistic regression to try and adjust for this. The Adjusted ORs and p values should be highlighted when reporting the differences between age groups throughout the abstract/manuscript.

Response: This has been done.

4. It was not clear whether there was variability in the dosing interval in the 21 individuals whose post-2nd dose data are included. The authors state that the 3 week interval changed to 12 weeks in the middle of the study. This is important as there may be significant difference in the post-boost antibody titre as has been seen for the ChAdox vaccine.

Response: in those with second dose data the dosing interval was 3 weeks. This has been clarified in the manuscript.

5. There are very few individuals with post 2nd dose samples (21) and stratifying further into age <80 for 80+ (5 vs 16) makes many statistical comparisons the authors have carried out meaningless. For example it is difficult to interpret the non-significant result in neutralising titres between the two age groups.

Response: we agree and can remove this analysis.

6. The authors state that they were interested in seeing if the 'poorer' responses persisted up to 12 weeks after the 1st dose. It is not clear whether the samples used to answer this question were longitudinal samples (i.e. repeat samples) taken from individuals who had samples taken earlier after vaccination. If it is not and these are different individuals, it would not be appropriate to use these data to compare 'early vs late responses' given the small sample size and inter-individual heterogeneity in antibody responses seen.

Response: these were not longitudinally collected samples. We will amend the text to clarify this point, though we do feel it is important to note that only 50% of people over 80 vaccinated 12 weeks prior have detectable neutralisation and are therefore vulnerable.

7. The authors state the “Age showed statistically significant correlation with serum neutralisation of WT virus after the first but not second dose”. Given the small sample size following the 2nd dose, this is not appropriate to conclude. The R is pretty much the same (-0.25 vs -0.29) but it is just that the p value is 0.28 potentially due to lack of power.

Response: *We will amend the text accordingly.*

8. Fig 1F - please make it clear what the red and blue neutralisation curves represent.

Response: *This has been clarified in figure legend.*

9. The authors state that “IgA concentrations was significantly correlated with age (Figure 2B).” It should be made clear if this is after 1 dose, which would be the most appropriate comparison. Though again the paucity of individuals at younger age groups makes this comparison difficult to interpret in the context of a correlation.

Response: *We agree and this has been clarified in both the text and legend. In addition, 37 individuals less than 50 years have been included in the analysis to bolster the data in the younger age group.*

10. The authors state that “IgG1-4 levels after dose 1 correlated strongly with neutralisation”. I do not think that an R of 0.27 and 0.4 (Fig 2C and D) can be called a ‘strong’ correlation.

Response: *We agree and have amended the text.*

11. There are 24 comparisons made in Figure 3B looking at skewing of V gene use. While in the statistical methods the authors state that Bonferroni correction was performed to adjust for multiple comparisons, it is not clear in this figure or legend that this was done. Please confirm if the p values presented are following Bonferroni adjustment as if not it is possible that at least some of the significant results would be due to chance.

Response: *We did not correct in this particular analysis but have now done so by using a Benjamini Hochberg FDR correction, setting the threshold at 0.1. This is clarified in the legend.*

Bonferroni adjustment was made for multiple comparisons in the linear correlation analyses between binding antibody levels, ID50, age and T cell responses. This has been clarified in the text.

12. The T-cell results section is labelled as “CD4 and CD8 T cell responses to spike following mRNA vaccination”. This is misleading as nowhere are responses stratified by CD4 and CD8. Given the differences in IL-2 and IFN-g across doses and age groups the authors report, the manuscript would benefit significantly if there were ICS data (or CD4/CD8 depleted ELISpot data) showing whether these differences were due to differences in spike-specific CD4 and CD8 induction across ages or doses. This is a major deficiency in my opinion.

Response: *we are in the process of obtaining CD4/CD8 depleted ELISpot data to fill this gap.*

13. There appears to be quite high background in the IFN γ assays – surprising that even in <80s, the assay used does not pick up many T-cell responses after a single dose, which is in contrast to some other studies. This may be due to some methodological issues.

Response: *We have looked further into this. If by background the reviewer is referring to spots from wells that just contain cells and media, this has been deducted from the reported levels in the tested samples.*

If this is reference to the levels in our unexposed group, it is worth noting that Li et al Nat Med <https://www.nature.com/articles/s41591-021-01330-9> reported IFN γ levels in their placebo group, which are comparable to that of our unexposed group. In our study, 8 of them are 0, 6 are less than 50 per million; there is a single high response at about 250 but as we did not see a technical difficulty with that and have left it in.

In Fig 3 Li et al show the T cell IFN gamma data is shown as spots per 10e5 while our is shown as per 10e6 as such the placebo group (most similar to our unexposed), and scaling the Li data to per 10e6 PBMC, it is approx a range of 5-100 SFU in the younger placebo group and 10-100 in the older group. The exceptions are three individuals at about 200-300 SFU and one at about 900. The Li et al paper also see an occasional high responder in this unexposed group.

-Fig 3 for vaccinated groups the T cell data is after the boost dose at 7 and 21days so should be compared to our second dose data the Li paper has this as 10-8000 SFU per 10e6 across the groups with a mean of about 500. We have reported 10-2000 and our mean at about 250

-Fig 3 CEF distribution in the Li paper is similar to the range in our dataset.

Taken together our data are not discordant with the Li et al findings and our methods robust.

a) In some cases significantly less PBMCs appear to have been used (100,000 cells/well). The responses seen with this may be significantly different and unable to detect low level responses compared to using 250,000 cells/well (a more usual number). This may be a significant methodological weakness – particularly if there was any bias in how many in each group were used with lower vs higher cell numbers.

Response -The Li paper used 100K per well to stimulate; 100k cells was our minimum.

b) There are no details on overlapping peptides for spike – whole spike? What length of peptides, what was the overlap?

Response:

c) Why did the authors choose 1mcg/ml as a final concentration to stimulate with in T-cell assays? Many assays/publications have used 2mcg/mL (or in some cases 10mcg/mL – eg the ChadOx vaccine immunogenicity data from clinical trials). This could have affected the magnitude of responses seen.

Response: The Li paper used 1mcg/ml which is also what the manufacturer recommended.

14. The age stratified statistical comparisons with the <80 post 2nd dose T-cell data seem a little meaningless with 5 data points.

Response: we agree and will now include additional data on participants who have now had their second dose vaccine.

15. I am not sure that the autoantibody data add much to this manuscript.

Response: we did this as auto antibodies have been associated with senescence. We also have data on inflammatory molecules, also noted to be raised in immune senescence. This analysis did not show differences either. Whilst negative data we feel that the investigation was warranted and have moved both to supplementary material. We hope that this is acceptable.

Referee #2:

Collier and colleagues report on a cohort of 101 individuals of median age 81 years that received 1 or 2 doses of SARS-CoV-2 mRNA vaccine (BTN162b2). The group was split into those that are 80 or older, and those that were younger than 80 for analysis. Samples from participants were analyzed for neutralizing activity against Wuhan and the B.1.17 variant, ELISA binding titer, V gene usage, somatic mutation and T cell cytokine responses. They find that the people that were over 80 had lower neut responses after 1 dose of vaccine but caught up after the second dose. That as many as 55-57.8% of the older individuals receiving a single vaccine had measurable neut titers against B.1.1.7. They performed bulk BCR sequencing and found some changes in VH representation between the 2 groups but these were not correlated to neut activity which would have required antibody cloning and expression. Small differences were seen in somatic mutation. T cell responses to spike were measured by INFg and IL2 spot assays.

This an important topic, and one that has already been reported on by others. The key result that responses are generally lower after one than after 2 immunizations is well established. Moreover, it has also been reported that (Refs3 and 19) older individuals are most likely to show only low or no detectable levels of neut activity after the first dose. Whether this correlates with protection from hospitalization, which is the key parameter, is not clear and is clarified by this report.

Response: Refs 3 and 19 are from clinical trials and do not represent real world data. Trial participants are often not representative of communities. Furthermore the numbers were very small in both those studies and no testing against variants or indeed D614G Spike virus was done. Therefore we have provided a significant amount of new and important data in this at risk population.

Overall, the analysis is superficial for both B and T cell responses. On the B cell side antibody cloning and expression would be required to further understand what happened and on the T cell side much broader analysis of phenotype and cytokine responses including single cell mRNA analysis would be important.

Response: we aimed to perform extensive B and T cell studies but given the elderly population we had to be pragmatic in terms of the amount of blood taken and prioritise the analyses accordingly. Our data are focused on likely protection from infection and we have data on variants that we would like to add.

Antibody cloning and expression samples only a small proportion of antigen-specific B cells and, while it does allow for functional interrogation of antibody function, it does not necessarily provide an adequate overview of wider B cell response which is results in serum antibody activity. Ideally, we would have combined antibody cloning, repertoire sequencing and serum evaluation but there were insufficient PBMC available. We elected to pursue BCR repertoire analysis as it provides a more in-depth picture of systemic alterations in the antibody response which could underpin this inferior serum response. We have now enhanced our analysis by considering SHM and ratios of coding versus non coding changes. In addition

we have done an analysis looking at public B cell responses and their presence in young and old participants.

We are doing additional experiments to demonstrate cytokine production by CD4 and CD8 T cells.

In addition, it is not at all clear why they separate individuals into 2 groups at age 80. Why not analyze on a continuum to see if there is an age cut off for the effects they report.

Response: We chose age 80 years as the cut-off for the age groups because of limited data on vaccine response in the elderly who are at the greatest risk from COVID-19. A public health response to controlling the pandemic in some countries including the UK has been to increase the dosing schedule from 3 to 12 weeks in order to vaccinate more people. An important question to address is if this approach will be effective in protecting the elderly. We have shown correlates for vaccine response which suggest that the elderly maybe at risk of inadequate protection from only one dose of the vaccine.

However, as the reviewer rightly points out, age is a continuum and so we presented linear correlation analyses of serum neutralisation ID50 and binding antibody levels with age as a continuous variable and these are presented in Figures 1D, 2B, Supp Figure 2A. We will analyse T cell responses by age as a continuous variable as well and present this. For policy purposes age cut-offs become important so we have kept this central to the work presented.

Referee #3:

The authors have put forth significant effort and the apparent intent of this report could be of great value.

The paper would benefit from some significant modifications. The (2-3) key points are difficult to determine from the abstract. The "study" design is not apparent in the abstract and even in the text it is confusing to follow.

The key points (whatever the authors determine to be key) should be clear in abstract, results, and then discussed, explained, and their significance explained in the discussion.

The discussion currently seems to ramble and include multiple only somewhat relevant facts (and some that seem less relevant), difficult to interpret implied implications for these findings, and the message is not clear or cohesive, so it is difficult to read and also to take away key messages.

Response: we will amend accordingly

Specific comments:

Please include data related to age as a continuous variable, an arbitrary cutoff of 80 years has minimal value to public health operations---if in fact a continuous variable analysis were used it may show that the drop off in response after 1st dose occurs closer to 60 or 70 years, or is more extreme at 85-90 years. Here the reader is left to wonder but certainly there is nothing magical about the immune response at the age of 80 as compared to 75 or 85.

Response: As elaborated on above we agree and will amend

Seems the key point of the paper is that older individuals do not mount a clinically significant response (at least if Neut Ab turns out to be the Correlate of Protection (CoP) for mRNA) after a single dose of mRNA. This is key and should be fine tuned in the message and results.

Response: we agree and will amend

It is also important for the authors to clarify to the reader that the CoP is unknown. It may not be neutralizing antibody and rather may be binding antibody---if so, the premise of this report is flawed. That unknown should be declared up front and again in the discussion very clearly.

Response: we agree and will amend to reflect this point though reports correlate neutralisation with protection rather than RBD specific antibodies.

As for T cell responses, describing the quality of response without context to the reader is not helpful. There is no known T cell mechanism of protection for these vaccines and thus the intent of the data here are even more speculative. The authors could elaborate on the meaning of the different types of T cell responses but clarify the value is is speculative for this vaccine/infection.

Response: Data such

as: https://papers.ssrn.com/sol3/papers.cfm?abstract_id=3796835 show an effect in preventing hospitalisations in older people even though antibody titres are low, suggesting T cells are important. We will add this to the discussion and clarify what is speculation.

Multiple speculative statements need to be removed or explained as such---related to speculation about the presumed response against the 351 variant or "escape variants". There is no data here and this is speculative.

The ref to high dose influenza is odd---is the author suggesting high dose SARS CoV2 vaccine is needed?

The ref to HIV vaccine responses is not useful at least as written---HIV vaccines have not been effective and so the

immune responses to ineffective investigational vaccines are not a benchmark for comparing to a known effective vaccine---this info and ref seems out of place and not useful.

Response: we agree and will amend. We now have included data on neutralisation activity against the variants P1 and B.1.135 and will discuss this.

The speculation in the last of the discussion that those who respond poorly to 1st injection (but do respond to 2nd) will have diminished durability is not founded in data here or elsewhere and should be removed as a speculation. Could be stated as a question to be answered.

Response: we agree and will amend

Overall, there is a lot of important data here, but the story and key messages are not quite there as written for a top tier journal publication intended for broad audience.

Response: we thank the reviewer and will amend to reflect this opinion.

Reviewer Reports on the First Revision:

N/A

Author Rebuttals to First Revision:

We have responded point by point to the reviewer comments below.

The authors present some of the only data available to date on SARS-CoV-2 mRNA vaccine immunogenicity in an elderly cohort (n=101), with a median age of 81 (IQR 70-84). The majority of data presented are following one dose of the BNT162b2 vaccine, with a smaller number (n=21) having received two doses. The primary findings are those comparing pseudovirus neutralisation titres (to both D614G and B.1.1.7 spike constructs) in those <80 years compared to those 80 and above; with a geometric mean neutralisation titre to D614G of 83.4 (95% CI 52.0 - 133.7) in those under 80 vs 46.6 (95% CI 33.5-64.8) in the 80+ group after one dose. The OR (adjusted for sex and time from the 1st dose) between groups is reported as 3.7 (1.2-11.2), p=0.02. Neutralisation titres in both age groups were found to be lower to B.1.1.7 than D614G spike pseudotyped viruses. The authors also used ex vivo IFN-gamma/IL-2 Fluorospot assays to interrogate differences in spike-specific T-cell response across age groups and first/second doses. The main findings are that the second dose results in a boost to IFN-gamma producing T-cells elicited after the first dose in both age groups (<80 and 80+), but no significant increase in IL-2 producing T-cells. Following the first vaccine dose, no difference in IFN-gamma T-cell response was seen in individuals <80 compared to those 80 and over, but a lower IL-2 T-cell response was seen in the 80+ age group. There are a number of other data presented of varying relevance, novelty and interest, including correlation of binding antibody responses with neutralisation titres, levels of somatic hypermutation and levels of autoantibodies (eg anti-MPO and anti-cardiolipin).

Overall, the authors should be commended for gathering data in an elderly cohort, as there are very few manuscripts with such data for SARS-CoV-2 vaccine responses (e.g. Anderson et al, NEJM, 2020 for mRNA-1273; Li et al Nature Medicine, 2021 for BNT162b1; Parry et al, 2021, SSRN preprint comparing BNT162b2 and ChAdOx1, as well as the subset in the Phase I Walsh et al study in NEJM). The Li et al study in Chinese adults should be referenced and was likely published after submission of the current manuscript. While it is using the BNT162b1 vaccine, it includes data from 72 adults aged 65-85, along with an appropriate

control group of adults aged 18-55 in the context of a double blind phase I immunogenicity study (live virus neutralisation, binding antibody and IFN-gamma T-cell responses).

Response: we thank the reviewer for appreciating the effort we have made to bring these data together, as suboptimal immune responses are likely to be a long term problem for control of SARS-CoV-2 and may promote emergence of novel hyper mutated variants. These data should drive the search for ways of boosting immunity following vaccination in the elderly. We have cited the papers suggested.

However, other than the pseudovirus neutralisation data which are robust and useful, other aspects of the manuscript either have some methodological or statistical concerns and/or feel like they have been added in to expand the data to a full manuscript without any particular utility or focused and thorough immunological exploration of what may underlie the poorer antibody responses observed. Specific comments are given below:

Response: we thank the reviewer for this comment and would like to add that we have also generated data on B.1.351 and P.1 viruses across age groups in order to make the work translatable to the evolving situation globally. These are the only data in the literature showing how age impacts neutralisation to variants. In addition the BCR analysis shows that elderly individuals have lower levels of SHM. We have now also added an analysis showing enrichment of public BCR clonotypes associated with neutralisation in younger vaccinees.

Finally our neutralisation, antibody and T cell analyses have been substantially expanded to include a larger younger group and in addition we have tested second dose samples from a total of 39 participants for T cell responses. As requested by the reviewer we have also tested T cell subsets for IL2 and IFNgamma responses to spike. We found that the IL2 and IFNgamma following spike peptide stimulation are coming from CD4 T cells and not CD8 cells. Anderson et al, NEJM 2020 had a similar finding for the Moderna 1273 mRNA vaccine; importantly our results are the only data we can find on CD4 v CD8 T cell responses for BNT162b2; the Li et al study on BNT162b1 used bulk PBMC. Finally our investigations on serum auto antibodies and inflammatory proteins have now been moved into the supplementary data. We believe the work presented is substantial and a significant advancement for the field.

1. The group presented that are <80 years old (n=42 of 101) have quite a wide age range and are still relatively old. Immunosenescence probably starts earlier than 80 years of age. To be a true representation of the title ('Age-related heterogeneity..'), the authors should ideally have included adequate numbers of participants <65 years, 65 - 79 and 80+ years.

Response: we agree and have generated additional data on a further 37 individuals in the <65 age group in the revised manuscript. There were however insufficient numbers across age bands to have three categories.

2. Are there any details on previous SARS-CoV-2 exposure in the cohort? I could not see that this was clearly presented and as several publications now show, prior infection results in a significantly higher antibody and T-cell response following the first dose of vaccine. Reassurance is needed that there is no imbalance in the % of individuals with prior exposure in groups (eg higher in younger age group), even if this is done based on anti-N antibodies in post-vaccine samples if pre-vaccine samples are not available. If this approach is used, then an extensively validated antibody assay used in clinical diagnostic laboratories should be utilised. There is of course the caveat about waning anti-N antibodies, so pre-vaccine samples are idea, but I appreciate this may not be available.

Response: we have data on N antibodies using a validated Luminex flow based assay that signifies prior exposure and have identified individuals with probable previous SARS-CoV-2 infection; 5 in the <80 years group and 5 in the ≥ 80 years group. We have now adjusted the analysis for prior infection as shown in Table 1 and Supplementary table 1.

3. There is considerable heterogeneity in the sampling period after the 1st dose (3 - 8 weeks), which Fig 1C demonstrates. The authors have rightly used logistic regression to try and adjust for this. The Adjusted ORs and p values should be highlighted when reporting the differences between age groups throughout the abstract/manuscript.

Response: We appreciate this comment and the changes have now been made. We have moved Fig 1C to supplementary material.

4. It was not clear whether there was variability in the dosing interval in the 21 individuals

whose post-2nd dose data are included. The authors state that the 3 week interval changed to 12 weeks in the middle of the study. This is important as there may be significant difference in the post-boost antibody titre as has been seen for the ChAdox vaccine.

Response: There were some individuals with second dose at 3 weeks post first dose and some at 12 weeks. We have post second dose data in a limited number and we found no difference between the two, with the caveat of small sample size (n=21 at 3 weeks and n=11 at 12 weeks). The figure below has been added to extended data figure 2.

5. There are very few individuals with post 2nd dose samples (21) and stratifying further into age <80 for 80+ (5 vs 16) makes many statistical comparisons the authors have carried out meaningless. For example it is difficult to interpret the non-significant result in neutralising titres between the two age groups.

Response: we have a greater number of post second dose results but agree with the reviewer that the sample size is too small to interpret the result.

6. The authors state that they were interested in seeing if the 'poorer' responses persisted up to 12 weeks after the 1st dose. It is not clear whether the samples used to answer this question were longitudinal samples (i.e. repeat samples) taken from individuals who had samples taken earlier after vaccination. If it is not and these are different individuals, it would not be appropriate to use these data to compare 'early vs late responses' given the small sample size and inter-individual heterogeneity in antibody responses seen.

Response: these were not longitudinally collected samples. We have amended the text to clarify this point, though we do feel it is important to note that when people over 80 vaccinated and presenting for the second dose 12 weeks later, 505 have no detectable neutralisation and are therefore have been vulnerable for those 12 weeks.

7. The authors state the “Age showed statistically significant correlation with serum neutralisation of WT virus after the first but not second dose”. Given the small sample size following the 2nd dose, this is not appropriate to conclude. The R is pretty much the same (-0.25 vs -0.29) but it is just that the p value is 0.28 potentially due to lack of power.

Response: This you for pointing this out. We have amended the text accordingly.

8. Fig 1F - please make it clear what the red and blue neutralisation curves represent.

Response: This has been clarified in figure legend.

9. The authors state that “IgA concentrations was significantly correlated with age (Figure 2B).” It should be made clear if this is after 1 dose, which would be the most appropriate comparison. Though again the paucity of individuals at younger age groups makes this comparison difficult to interpret in the context of a correlation.

Response: We agree and this has been clarified in both the text and legend. In addition, 37 individuals less than 50 years have been included in the analysis to bolster the data in the younger age group.

10. The authors state that “IgG1-4 levels after dose 1 correlated strongly with neutralisation”. I do not think that an R of 0.27 and 0.4 (Fig 2C and D) can be called a ‘strong’ correlation.

Response: We agree and have amended the text accordingly.

11. There are 24 comparisons made in Figure 3B looking at skewing of V gene use. While in the statistical methods the authors state that Bonferroni correction was performed to adjust for multiple comparisons, it is not clear in this figure or legend that this was done. Please

confirm if the p values presented are following Bonferroni adjustment as if not it is possible that at least some of the significant results would be due to chance.

Response: We did not correct for multiple comparisons in this particular analysis but have now done so by using a Benjamini Hochberg FDR correction, setting the threshold at 0.1. This is clarified in the legend. Bonferroni adjustment was made for multiple comparisons in the linear correlation analyses between binding antibody levels, ID50, age and T cell responses.

12. The T-cell results section is labelled as “CD4 and CD8 T cell responses to spike following mRNA vaccination”. This is misleading as nowhere are responses stratified by CD4 and CD8. Given the differences in IL-2 and IFN-g across doses and age groups the authors report, the manuscript would benefit significantly if there were ICS data (or CD4/CD8 depleted ELISpot data) showing whether these differences were due to differences in spike-specific CD4 and CD8 induction across ages or doses. This is a major deficiency in my opinion.

Response: we have obtained data for CD4/CD8 subsets by PBMC depletion as suggested by the review in a subsample of participants. The results show interestingly that IL2 responses and IFN γ responses are primarily from CD4, similar to findings from Moderna 1273 (Anderson et al) and ChAdOx1 (Ewer et al, Nat Med). To our knowledge there are no data on BNT162b2.

13. There appears to be quite high background in the IFN γ assays – surprising that even in <80s, the assay used does not pick up many T-cell responses after a single dose, which is in contrast to some other studies. This may be due to some methodological issues.

Response: We have looked further into this. If by background the reviewer is referring to spots from wells that just contain cells and media, this has been deducted from the reported levels in the tested samples.

If this is reference to the levels in our unexposed group, it is worth noting that Li et al Nat Med <https://www.nature.com/articles/s41591-021-01330-9> reported IFN γ levels in their placebo group, which are comparable to that of our unexposed group. In our study, 8 of them are 0, 6 are less than 50 per million; there is a single high response at about 250 but as we did not see a technical difficulty with that and have left it in.

In Fig 3 Li et al show the T cell IFN gamma data is shown as spots per 10e5 while our is shown as per 10e6 as such the placebo group (most similar to our unexposed), and scaling the Li data to per 10e6 PBMC, it is approx a range of 5-100 SFU in the younger placebo group and 10-100 in the older group. The exceptions are three individuals at about 200- 300 SFU and one at about 900. The Li et al paper also see an occasional high responder in this unexposed group.

In the Li et al paper Fig 3, the T cell data are those found after the vaccine boost dose at 7 and 21 days. This should be compared to our second dose data. The Li paper reports 10-8000 SFU per 10e6 across the groups with a mean of about 500. We have reported 10-2000 and our mean is at about 250.

-Fig 3 CEF distribution in the Li paper is similar to the range in our dataset.

- Taken together our data are not discordant with the Li et al findings and our methods robust.

- a) In some cases significantly less PBMCs appear to have been used (100,000 cells/well). The responses seen with this may be significantly different and unable to detect low level responses compared to using 250,000 cells/well (a more usual number). This may be a significant methodological weakness – particularly if there was any bias in how many in each group were used with lower vs higher cell numbers.

Response: The Li paper used 100K per well to stimulate; we have now also used 100k cells in our study as a minimum.

- b) There are no details on overlapping peptides for spike – whole spike? What length of

peptides, what was the overlap?

Response: The peptides are 15aa amino acids with 11 amino acid overlaps across the whole of spike (<https://www.miltenyibiotec.com/GB-en/products/peptivator-sars-cov-2-prot-s-107090.html#gref>). We have added the details of peptides used to the methods.

c) Why did the authors choose 1mcg/ml as a final concentration to stimulate with in T-cell assays? Many assays/publications have used 2mcg/mL (or in some cases 10mcg/mL – eg the ChadOx vaccine immunogenicity data from clinical trials). This could have affected the magnitude of responses seen.

Response: *The Li paper used 1mcg/ml which is also what the manufacturer recommended.*

14. The age stratified statistical comparisons with the <80 post 2nd dose T-cell data seem a little meaningless with 5 data points.

Response: *we agree and have now included additional data on participants following second dose*

15. I am not sure that the autoantibody data add much to this manuscript.

Response: *we did this as auto antibodies have been associated with immune senescence. We also now have data on a large panel of molecules associated with immune senescence. This analysis did not show differences either. Whilst negative data we feel that the investigation was warranted and have moved both to supplementary material. We hope that this is acceptable, though would be happy to remove it entirely if asked to do so.*

Referee #2:

Collier and colleagues report on a cohort of 101 individuals of median age 81 years that received 1 or 2 doses of SARS-CoV-2 mRNA vaccine (BTN162b2). The group was split into those that are 80 or older, and those that were younger than 80 for analysis. Samples from participants were analyzed for neutralizing activity against Wuhan and the B.1.17 variant, ELISA binding titer, V gene usage, somatic mutation and T cell cytokine responses. They find that the people that were over 80 had lower neut responses after 1 dose of vaccine but

caught up after the second dose. That as many as 55-57.8% of the older individuals receiving a single vaccine had measurable neut titers against B.1.1.7. They performed bulk BCR sequencing and found some changes in VH representation between the 2 groups but these were not correlated to neut activity which would have required antibody cloning and expression. Small differences were seen in somatic mutation. T cell responses to spike were measured by INFg and IL2 spot assays.

This an important topic, and one that has already been reported on by others. The key result that responses are generally lower after one than after 2 immunizations is well established. Moreover, it has also been reported that (Refs3 and 19) older individuals are most likely to show only low or no detectable levels of neut activity after the first dose. Whether this correlates with protection from hospitalization, which is the key parameter, is not clear and is clarified by this report.

Response: We thank the reviewer for this comment and agree. Refs 3 and 19 from phase I mRNA clinical trials with small numbers of participants reported low neutralising antibody responses after the first dose in the elderly. We have provided real world data in significant numbers of individuals, and importantly tested variants to understand what protection against current strains might be in this at risk population.

Overall, the analysis is superficial for both B and T cell responses. On the B cell side antibody cloning and expression would be required to further understand what happened and on the T cell side much broader analysis of phenotype and cytokine responses including single cell mRNA analysis would be important.

Response: Our data are focused on implications of age on likely protection from infection following one dose of vaccine, pertinent given the dosing interval is being increased from 1-4 months or more across the world due to shortages. Our large dataset on neutralisation of variants of concern make our work highly relevant to current and future control efforts. We believe we have made considerable progress in exploring the correlates of suboptimal responses within this overall context.

Antibody cloning and expression samples only a small proportion of antigen-specific B cells and, while it does allow for functional interrogation of antibody function, it does not necessarily provide an adequate overview of wider B cell response which results in serum antibody activity. Ideally, we would have combined antibody cloning, repertoire sequencing and serum evaluation but there were insufficient PBMC available. We elected to pursue BCR repertoire analysis as it provides a more in-depth picture of systemic alterations in the antibody response which could underpin this inferior serum response. We have now enhanced our analysis by considering public B cell responses and their presence in our participants. We have also performed flow cytometry in a subset of individuals to quantify B memory and spike specific B cells and relate this to neutralisation.

In terms of T cell analysis, our additional experiments to demonstrate cytokine production by CD4 and CD8 T cells have been highly informative showing that IFN γ and IL2 cytokines following vaccination are largely CD4 T cell derived. Single cell mRNA analysis was not feasible in the time frame available, and we feel also beyond the scope of the paper given little support for T cells protecting against initial infection (Reviewer 3's comments are pertinent here). We would like to explore T cell help for high affinity antibody production in future.

In addition, it is not at all clear why they separate individuals into 2 groups at age 80. Why not analyze on a continuum to see if there is an age cut off for the effects they report.

Response: We thank the reviewer for this comment. As now shown in Figure 1 there is a drop off in the proportions of individuals with effective neutralisation (ID₅₀>20) around 80 years of age, and a similar picture for cytokine responses in PBMC. For policy purposes age cut-offs are important and given the drop off in T cell responses in the over 80 year olds observed in our work and others (Parry et al, SSRN 2021, <https://dx.doi.org/10.2139/ssrn.3816840>), we have kept this central to the work presented. We have nonetheless also presented linear regression analyses of serum neutralisation ID₅₀ and binding antibody levels with age as a continuous variable and these are presented. We have also analysed T cell responses by age as a continuous variable as well.

Referee #3:

The authors have put forth significant effort and the apparent intent of this report could be of

great value.

The paper would benefit from some significant modifications. The (2-3) key points are difficult to determine from the abstract. The “study” design is not apparent in the abstract and even in the text it is confusing to follow.

The key points (whatever the authors determine to be key) should be clear in abstract, results, and then discussed, explained, and their significance explained in the discussion.

The discussion currently seems to ramble and include multiple only somewhat relevant facts (and some that seem less relevant), difficult to interpret implied implications for these findings, and the message is not clear or cohesive, so it is difficult to read and also to take away key messages.

Response: we thank the reviewer for these comments and have now amended the abstract, intro and discussion to read more clearly and cohesively.

Specific comments:

Please include data related to age as a continuous variable, an arbitrary cutoff of 80 years has minimal value to public health operations---if in fact a continuous variable analysis were used it may show that the drop off in response after 1st dose occurs closer to 60 or 70 years, or is more extreme at 85-90 years. Here the reader is left to wonder but certainly there is nothing magical about the immune response at the age of 80 as compared to 75 or 85.

Response: we thank the reviewer for these comments and have now shown as much of the data as a continuous variable as possible. Of note, we observe a drop off in both neutralisation and T cell responses at around 80 and therefore have also kept the analyses comparing above and below 80 year olds. Age cut offs are used frequently in public health, for example seasonal influenza vaccination of over 65 year olds in the UK.

Seems the key point of the paper is that older individuals do not mount a clinically significant response (at least if Neut Ab turns out to be the Correlate of Protection (CoP) for mRNA) after a single dose of mRNA. This is key and should be fine tuned in the message and results.

Response: since the submission two further papers have emerged identifying Neutralising Ab as a correlate of protection, one in human vaccinees from clinical trial data and one in

mice. We have now tried to emphasise more clearly that older individuals do not mount a clinically significant response.

It is also important for the authors to clarify to the reader that the CoP is unknown. It may not be neutralizing antibody and rather may be binding antibody---if so, the premise of this report is flawed. That unknown should be declared up front and again in the discussion very clearly.

Response: we agree and have amended the text to reflect this point. Neutralisation has gained acceptance as a CoP in our view since the initial submission with some key papers. We have stated this position whilst recognising that T cells may have a role in preventing / limiting disease progression.

As for T cell responses, describing the quality of response without context to the reader is not helpful. There is no known T cell mechanism of protection for these vaccines and thus the intent of the data here are even more speculative. The authors could elaborate on the meaning of the different types of T cell responses but clarify the value is is speculative for this vaccine/infection.

Response: We have now provided some context for including T cell data, and mentioned where we are speculating in the discussion. For context we have added the paper from McMahan K, et al. Nature. 2021;590(7847), showing that CD8 depletion in NHPs leads to lack of protection from challenge in convalescent macaques.

Multiple speculative statements need to be removed or explained as such---related to speculation about the presumed response against the 351 variant or “escape variants”. There is no data here and this is speculative.

The ref to high dose influenza is odd---is the author suggesting high dose SARS CoV2 vaccine is needed?

The ref to HIV vaccine responses is not useful at least as written---HIV vaccines have not been effective and so the immune responses to ineffective investigational vaccines are not a benchmark for comparing to a known effective vaccine---this info and ref seems out of place and not useful.

Response: We now have included data on neutralisation activity against the variants P1 and B.1.135 and have removed the sections identified by the reviewer.

The speculation in the last of the discussion that those who respond poorly to 1st injection (but do respond to 2nd) will have diminished durability is not founded in data here or elsewhere and should be removed as a speculation. Could be stated as a question to be answered.

Response: we agree and have removed this sentence.

Overall, there is a lot of important data here, but the story and key messages are not quite there as written for a top tier journal publication intended for broad audience.

Response: we thank the reviewer and hope that our revised paper is acceptable.

We look forward to hearing from you

Sincerely

Ravi Gupta (on behalf of the authors)

Reviewer Reports on the Second Revision:

Referees' comments:

Referee #2 (Remarks to the Author):

The abstract does not adequately point out that the elderly catch up after the second dose. This is very clearly written in the conclusion and should be added to the abstract.

The paragraph on variants of concern, starting on line 120 should also include data on people that received 2 doses of vaccine. It is otherwise incomplete and for public policy it would be important to emphasize that 2 doses corrects the problem if in fact it does as might be predicted from the data in Fig. 1b.

Fig. 1d should be supplementary

Referee #3 (Remarks to the Author):

The paper has been strengthened by the improvements made.

The Correlate of protection induced by mRNA vaccine has not been demonstrated---this report is pending in the coming weeks and we are all anxiously awaiting the news.

The Khoury et al paper cited (#4 ref) implies this as per natural infection but the data are just not settled for CoP from mRNA or any vaccination.

Thus, the authors should more carefully and accurately describe their interpretation of the possible CoP for vaccine response. It is still written as though it is certain to be NAb.

It would be very untimely if NAb were not shown to be the CoP in the coming weeks and these author statements were inaccurate.

Author Rebuttals to Second Revision:

Reviewer 2

The abstract does not adequately point out that the elderly catch up after the second dose. This is very clearly written in the conclusion and should be added to the abstract.

Response: we have now amended the abstract

The paragraph on variants of concern, starting on line 120 should also include data on people that received 2 doses of vaccine. It is otherwise incomplete and for public policy it would be important to emphasize that 2 doses corrects the problem if in fact it does as might be predicted from the data in Fig. 1b.

Response: we have now addressed this

Fig. 1d should be supplementary

Response: We have moved this to supplementary

Reviewer 3

The paper has been strengthened by the improvements made.

Response: we thank the reviewer for this comment

The Correlate of protection induced by mRNA vaccine has not been demonstrated---this report is pending in the coming weeks and we are all anxiously awaiting the news.

Response: we agree and have toned down the text to reflect this

The Khoury et al paper cited (#4 ref) implies this as per natural infection but the data are just not settled for CoP from mRNA or any vaccination.

Response: we agree and have toned down the text to reflect this

Thus, the authors should more carefully and accurately describe their interpretation of the possible CoP for vaccine response. It is still written as though it is certain to be NAb.

It would be very untimely if NAb were not shown to be the CoP in the coming weeks and these author statements were inaccurate.

Response: we agree and have made changes highlighted in yellow

We look forward to hearing from you

Sincerely

Ravi Gupta (on behalf of the authors)